# Scope: A Self-supervised Framework for Improving Faithfulness in Conditional Text Generation

**Song Duong**[*,1,4]    **Florian Le Bronnec**[*,1,2]    **Alexandre Allauzen**[2]    **Vincent Guigue**[3]
**Alberto Lumbreras**[4]    **Laure Soulier**[1]    **Patrick Gallinari**[1,4]

[1]Sorbonne Université, CNRS, ISIR, F-75005 Paris, France
[2]Miles Team, LAMSADE, Université Paris-Dauphine, Université PSL, CNRS, 75016 Paris, France
[3]AgroParisTech, UMR MIA-PS, Palaiseau, France
[4]Criteo AI Lab, Paris, France

## Abstract

Large Language Models (LLMs), when used for conditional text generation, often produce hallucinations, i.e., information that is unfaithful or not grounded in the input context. This issue arises in typical conditional text generation tasks, such as text summarization and data-to-text generation, where the goal is to produce fluent text based on contextual input. When fine-tuned on specific domains, LLMs struggle to provide faithful answers to a given context, often adding information or generating errors. One underlying cause of this issue is that LLMs rely on statistical patterns learned from their training data. This reliance can interfere with the model's ability to stay faithful to a provided context, leading to the generation of ungrounded information. We build upon this observation and introduce a novel self-supervised method for generating a training set of unfaithful samples. We then refine the model using a training process that encourages the generation of grounded outputs over unfaithful ones, drawing on preference-based training. Our approach leads to significantly more grounded text generation, outperforming existing self-supervised techniques in faithfulness, as evaluated through automatic metrics, LLM-based assessments, and human evaluations. Code is available at https://github.com/sngdng/scope-faithfulness.

## 1 Introduction

Large Language Models (LLMs) are widely used for generating fluent and coherent text completions based on input contexts (Brown et al., 2020). These models generate completions by leveraging the statistical patterns encoded in their parameters, which are learned from extensive training data. While these parameters provide the model with a broad knowledge of various topics, they can also cause interference. This occurs when the model combines information provided in the input context with general patterns from its training data, potentially leading to inaccuracies. More generally, irrelevant content generated by a LLM is commonly referred to as *hallucinations* (Rebuffel et al., 2022; Maynez et al., 2020). To mitigate these hallucinations, two primary dimensions are considered: **factuality** and **faithfulness** (Huang et al., 2023). Factuality refers to whether the model's generated information aligns with external, real-world knowledge and is typically evaluated against a reference dataset or established knowledge. Faithfulness, on the other hand, evaluates how accurately the generated content reflects the information provided in the input context. A model may produce factual but unfaithful content if, while true with respect to world knowledge, it distorts important details from the input or adds extra information (see Table 1). This is particularly crucial in fields where accurate information transfer is essential. For instance, in medical transcription, the text output must accurately reflect the content of the medical record without introducing any distortions (Cawsey et al., 1997).

In this paper, we focus on the generation of **faithful** responses grounded in a self-contained input context. A major challenge concerning faithfulness is the difficulty of annotating data and there is no standard way to determine if a text is faithful to an input context. As a result, annotation is typically performed by humans (Goyal & Durrett, 2021; Kryscinski et al., 2020). However, this approach is

---

[*]Equal contribution. Corresponding authors: s.duong@criteo.com, florian.le-bronnec@dauphine.psl.eu

**Patient Data (Input)**:

| Age | Sex | Symptoms | Diagnosis | Treatment |
|-----|-----|----------|-----------|-----------|
| 45 | Male | Persistent cough | Pneumonia | Antibiotics |

**Output Examples**:

| Faithful | Factful | Output |
|----------|---------|--------|
| No | No | 21 y.o. female with a headache due to a migraine is given antibiotics. |
| No | Yes | 45 y.o. male with a cough due to pneumonia is given amoxicillin. |
| Yes | Yes | 45 y.o. male with a cough due to pneumonia is given antibiotics. |

Table 1: An example of faithful and factful combinations in LLM for data-to-text generation in a medical context. Unfaithful spans are highlighted in red. While amoxicillin is a common antibiotic prescription for pneumonia, the name of the antibiotics is not the mentioned in the table.

costly, not scalable, and the resulting annotations might not transfer to other domains. To circumvent the lack of annotated data, some unsupervised methods have been proposed. A first line of research consists of leveraging a contrastive loss on hidden representations (Zhao et al., 2020; Kryscinski et al., 2019). These methods have demonstrated improvements on small models (around 500 million parameters), but they have not yet been benchmarked on recent LLMs. Our evaluations indicate that their effectiveness does not appear to extend to these larger models. Another direction consists of altering the decoding process of pre-trained models (Shi et al., 2023; van der Poel et al., 2022). While these methods work well on generalist text datasets, we found that on more domain specific tasks where a heavy fine-tuning is required such as data-to-text generation, these methods struggle to improve over standard fine-tuning of models (see Section 5).

Acknowledging these limitations, we propose a novel fine-tuning framework, tailored for recent LLMs. Drawing inspiration from recent work (Rafailov et al., 2023), we propose a method tailored for recent LLMs that trains the model to disfavor ungrounded generations. Unlike typical preference-tuning which involves human annotation of model-generated outputs, we aim for a self-supervised process to generate a dataset of *prefered* and *dispreferred* samples. Here, in the context of faithfulness, the goal is to teach the model to prefer the context-grounded reference labels over unfaithful ones that present hallucinations. A challenge then lies in the generation of representative unfaithful examples that convey effective learning signals. These examples should closely resemble target sentences while exhibiting realistic hallucinations. In conditional text generation tasks, hallucinations occur when the model's internal knowledge improperly influences the generation process (Maynez et al., 2020). Building on this observation, we propose an original procedure for automatically generating realistic examples. This generation process is fully unsupervised and does not require external resources. We apply our method to six datasets across various domains for data-to-text generation and text summarization. Data-to-text generation (Lin et al., 2024) involves converting structured data like tables into coherent language, while summarization condenses longer texts while preserving key information. Faithfulness is essential for both tasks to ensure the generated text accurately reflects the input data. To summarize, in this paper:

- We introduce SCOPE, a new method that leverages ideas from preference training by using a self-supervised generated dataset. In this approach, the model is trained to favor reference labels over carefully generated unfaithful samples.

- We empirically show that our approach significantly enhances the faithfulness of text generated by fine-tuned LLMs, surpassing current faithfulness-enhanced methods for conditional text generation.

- We bring new insights on the behavior of preference-tuning by analyzing its sensitivity to the effect of negative samples.

Our experiments reveal that training using SCOPE achieves up to a 14% improvement in faithfulness metrics over existing methods, according to automatic evaluation metrics. Furthermore, evaluations by both GPT-4 and human judges indicate that the generations with SCOPE are substantially more faithful, with an improved preference win rate against the supervised fine-tuned model that is in average 2.1 times higher than the baselines.

## 2 RELATED WORK

This section reviews methods for improving LLMs' faithfulness to input contexts. We focus exclusively on approaches that ensure generated content remains grounded in the provided information, excluding techniques related to factuality or external knowledge alignment.

**Faithfulness enhancement.** Several methods have been used for improving faithfulness of text summarization. A first line of work consist in using external tools to retrieve key entities or facts form the source document and use these as weak labels during training (Zhang et al., 2022). Chen et al. (2022) identify key entities using a Question-Answering system and modify the architecture of an encoder-decoder model to put more cross-attention weight on these entities. Zhu et al. (2021) propose to improve the faithfulness of summaries by extracting a knowledge graph from the input texts and embed it in the model cross-attention using a graph-transformer. Another line of work focuses on post-training improvements by bootstrapping model-generated outputs ranked by quality (Zhao et al., 2023; Liu et al., 2022; Zablotskaia et al., 2023). Regarding data-to-text generation, Rebuffel et al. (2022) propose a custom model architecture to reduce the effect of loosely aligned datasets, using token-level annotations and a multi-branch decoder model. The closest work to ours is from (Cao & Wang, 2021) which proposes a contrastive learning approach where synthetic samples are constructed using different tools like Named Entity Recognition (NER) models and back-translation. These approaches address specific forms of unfaithfulness and rely heavily on external tools such as NER or QA models, and are especially tailored for text summarization, while we target a more general focus. More recently, simpler methods that leverage only a pre-trained model have been proposed for summarization. Shi et al. (2023); van der Poel et al. (2022) downweight the probabilities of tokens that are not grounded in the input context, using an auxiliary LM without access to the input context. Lango & Dusek (2023); Zhang et al. (2023) both train a self-supervised classification model to detect hallucinations that can be used to guide the decoding process toward more faithful outputs. Tian et al. (2020) propose a method to estimate the decoder's confidence by analyzing cross-attention weights, encouraging greater focus on the source during generation. Our method focuses on a decoder-only architecture and uses a single model, providing a streamlined and efficient approach specifically tailored for general conditional text generation tasks without the need for complex external tools.

**Faithfulness evaluation.** Measuring faithfulness automatically is not straightforward. Traditional conditional text generation evaluation often relies on comparing the generated output to a reference text, typically measured using n-gram based metrics such as BLEU (Papineni et al., 2002) or ROUGE (Lin, 2004). However, reference-based metrics limitations are well known to correlate poorly with faithfulness (Fabbri et al., 2021; Gabriel et al., 2021). Both for summarization and data-to-text generation, new metrics evaluating the generation exclusively against the input context have been proposed, using QA models (Rebuffel et al., 2021; Scialom et al., 2021) or entity-matching metrics (Nan et al., 2021). In this work, we evaluate primarily our models using recent NLI-related metrics (Zha et al., 2023; Dušek & Kasner, 2020), and LLM-as-a-judge, focusing on faithfulness (Chiang & Lee, 2023; Gilardi et al., 2023). For data-to-text generation, we also report the PARENT metric (Dhingra et al., 2019), which computes n-gram overlap against elements of the source table cells.

**Preference tuning.** Recent instruction-tuned LLMs are often further refined through "human-feedback alignment" (Guo et al., 2024). These methods utilize human-crafted preference datasets, consisting of pairs of preferred and dispreferred texts $(y^+, y^-)$, typically obtained by collecting human feedback and ranking responses via voting. Recent work (Chen et al., 2024) uses the model's previous predictions in a self-play manner to iteratively improve the performance of chat-based models. Whether through an auxiliary preference model (Ziegler et al., 2019) or by directly tuning the models on the pairs (Rafailov et al., 2023), these approaches have demonstrated remarkable results in chat-based models. Our method leverages a preference framework without the need for human intervention and is specifically tailored for models trained on conditional text generation tasks.

## 3 METHOD

We introduce SCOPE, a novel approach designed to address hallucinations by overcoming the limitations of standard fine-tuning (Maynez et al., 2020; Cao & Wang, 2021). Unlike traditional methods, our two-stage process aims to enhance the model's faithfulness. In the first stage, we perform standard fine-tuning to initialize the model. In the second stage, we apply preference tuning, where the model is further optimized using synthetic samples that guide it toward generating more faithful outputs. An illustration of the method is presented on Figure 1.

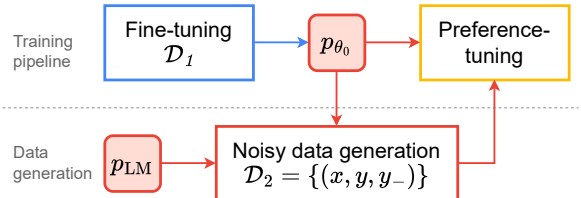

Figure 1: SCOPE training framework. A pre-trained model $p_{\text{LM}}$ is first fine-tuned on a subset $\mathcal{D}_1$ of $\mathcal{D}$ and produces a model $p_{\theta_0}$. A mixture of $p_{\text{LM}}$ and $p_{\theta_0}$ is then used to generate a synthetic preference dataset, which finally serves for preference fine-tuning.

## 3.1 TRAINING PHASE

Let $\mathcal{D} = (c_i, y_i)_{i=1}^N$ be an aligned dataset of context-target pairs used for training.

**Fine-tuning.** For the first stage, our goal is to get an initial version of a fine-tuned model. We start from a pre-trained model $p_{\text{LM}}$. To better leverage training samples, we propose for this part to train $p_{\text{LM}}$ only on the first half $\mathcal{D}_1$ of the samples of $\mathcal{D}$. We keep the second part $\mathcal{D}_2$ of the dataset for the next step. We denote by $p_{\theta_0}$ the model fine-tuned from $p_{\text{LM}}$ on $\mathcal{D}_1$ using cross-entropy. Given the strong sample efficiency of recent LLMs, we empirically found that for the datasets used, fine-tuning on only half of the samples was sufficient to achieve a strong initialization for the subsequent stage, see Appendix A.3.

**Preference-tuning.** The second phase involves contrastive learning. Training will be conducted on $\mathcal{D}_2$, the second half of the samples. We augment $\mathcal{D}_2$ with artificial unfaithful samples to get a dataset $\mathcal{D}_2 = \{c, y_i, y_i^-)\}_{i=1}^N$. Our complete process to generate these samples is described in Section 3.2. For each annotated target $y$, we have a corresponding noisy $y^-$ which contains unfaithful patterns.

While other baselines propose to use a custom contrastive loss often based on embeddings similarity, we propose optimizing the model to prefer $y$ over $y^-$ by leveraging the recent framework of preference tuning (Rafailov et al., 2023), with the following loss:

$$\mathcal{L}_\theta = -\mathbb{E}_{(c,y,y^-)\sim\mathcal{D}_2}\left[\log\sigma\left(\beta\log\frac{p_\theta(y\mid c)}{p_{\theta_0}(y\mid c)} - \beta\log\frac{p_\theta(y^-\mid c)}{p_{\theta_0}(y^-\mid c)}\right)\right], \qquad (1)$$

where $\sigma$ is the sigmoid function and $\beta$ is a scalar hyperparameter that quantifies how much $p_\theta$ deviates from $p_{\theta_0}$. Intuitively, minimizing $\mathcal{L}_\theta$ w.r.t. $\theta$ amounts to increasing the gap between the likelihood of generating grounded responses $y$ and non-grounded ones $y^-$. More details about the training dynamics can be found in Section 6. Additionally, we experimented with an alternative preference loss in Appendix A.5 and observed similar behavior.

## 3.2 UNFAITHFUL DATASET GENERATION

In this section, we present our method to generate unfaithful samples. Contrarily to other methods that rely on external tool, such a named entity recognition or number entity recognition, we propose an easier and more general method. When a LLM generates ungrounded spans of text, it is often caused by an interference between the context and the learned statistical patterns acquired during training. A *convincing unfaithful sample* generated by a LLM should satisfy at least two desiderata: **(i)** attain the same level of fluency than the target LLM, and **(ii)** being more or less consistent with the input while containing one or several spans of text not grounded in the input context. An ideal method would be to run our initial fine-tuned model $p_{\theta_0}$ and find among the samples the ones that are unfaithful. But as discussed in Section 1, accurately detecting unfaithful samples automatically is a difficult problem. Instead, we propose a simple unsupervised method to simulate the creation of noisy samples. Our strategy is to "force" the model to leak its internal statistical knowledge in the generation by adopting a noisy decoding method using two models simultaneously.

- The main model is $p_{\theta_0}$, the initially fine-tuned model on half the dataset. This model generates samples conditionally to the input context, $y \sim p_{\theta_0}(\cdot \mid c)$. It is supposed to generate text that is grounded in the input context, but can still contain inaccuracies due to its shortened training.

- The second model is $p_{\mathrm{LM}}$, the pre-trained counterpart of $p_{\theta_0}$. This model won't be given access to the input context and will simply sample from its context-free distribution $y \sim p_{\mathrm{LM}}(\cdot)$, generating general patterns that it has learned.

Both distributions are *de facto* fluent, but used individually might not be enough to teach anything during preference tuning. $p_{\mathrm{LM}}$ samples will obviously not be challenging enough, while $p_{\theta_0}$ samples won't contain enough hallucination patterns. Instead we propose to combine both during the decoding process. We generate these noisy samples token by token by sampling mainly from the grounded $p_{\theta_0}(\cdot \mid c)$ and randomly from the non-grounded $p_{\mathrm{LM}}$. This method introduces fluent but non-grounded artifacts, exhibiting both intrinsic errors, i.e., generated outputs that contradict the data, and extrinsic hallucinations, i.e., generated outputs that cannot be inferred from the data alone (see Table 18). Refer to Algorithm 1 for the complete details of the algorithm.

---

**Algorithm 1:** `noisy_generation`$(c, p_{\mathrm{LM}}, p_{\theta_0})$

---

**Input :** $c$ an input context, $p_{\mathrm{LM}}$ the pre-trained model, $p_{\theta_0}$ the fine-tuned model on $\mathcal{D}_1$.

**for** *token decoding step $t > 0$* **do**

    1. Sample: $\alpha_t \sim \mathrm{Bernoulli}(\alpha)$ ($\alpha_t \in \{0, 1\}$).

    2. Sample:

$$y_t^- \sim (1 - \alpha_t)p_{\theta_0}(\cdot \mid y_{<t}^-, c) + \alpha_t p_{\mathrm{LM}}(\cdot \mid y_{<t}^-) \tag{2}$$

**return** $y^-$;

---

The mixture is parameterized by $\alpha$, which tunes the noise level within the samples. $\alpha = 0$ corresponds to the fine-tuned model $p_{\theta_0}$ and $\alpha = 1$ corresponds to the unconditional model $p_{\mathrm{LM}}$. This parameter actually plays an important role: the noisy $y^-$ should contain divergences from the context but still be close enough to the true $y$ to provide a meaningful learning signal. This is a sensible step for preference learning, as illustrated later in the experiments (Section 6).

Our detailed pipeline is described in Algorithm 2. Existing preference tuning methods usually depend on offline preference data gathered from various sources and ranked through voting. In contrast, the originality of our approach lies in its ability to automatically generate unfaithful responses, simulating potential hallucinations from the model's internal state *without requiring supervision*. This distinguishes it from traditional preference training, which typically involves human intervention.

---

**Algorithm 2:** SCOPE (Self-supervised Context Preference).

---

**Input :** $\mathcal{D}$ the training data and $p_{\mathrm{LM}}$ the pre-trained model.

```
// Split the train data
```
$\mathcal{D}_1, \mathcal{D}_2 \leftarrow$ Split $\mathcal{D}$ into two halves

```
// 1. Initial fine-tuning
```
$p_{\theta_0} \leftarrow$ Fine-tune $p_{\mathrm{LM}}$ on $\mathcal{D}_1$

```
// 2. Noisy generation
```
$\widetilde{\mathcal{D}}_2 \leftarrow \{\}$

**for** $(c, y)$ *in* $\mathcal{D}_2$ **do**

    $y^- \leftarrow$ `noisy_generation`$(c, p_{\mathrm{LM}}, p_{\theta_0})$

    Append $(c, y, y^-)$ to $\widetilde{\mathcal{D}}_2$

```
// 3. Preference fine-tuning by optimizing Equation (1)
```
$p_\theta \leftarrow$ Preference fine-tune $p_{\theta_0}$ over $\widetilde{\mathcal{D}}_2$, using $y$ as the preferred label and $y^-$ as the negative example

**return** $p_\theta$;

---

## 4 EXPERIMENTS

### 4.1 TASKS AND DATASETS

We evaluate our method SCOPE on a total of 6 datasets, spanning multiple domains and difficulties, where generating grounded context is a crucial requirement. We first run experiments on four data-to-text generation datasets. **ToTTo** (Parikh et al., 2020) is an English dataset with Wikipedia tables

where specific cells are highlighted, paired with a sentence describing those cells. **WebNLG 2020 (English)** (Castro Ferreira et al., 2020) is an English dataset composed of pairs of knowledge graphs and text crawled from DBpedia. **E2E** (Dušek et al., 2019) is an English benchmark dataset that verbalizes key-value attribute pairs in the restaurant domain. **FeTaQA** (Nan et al., 2022) is an English table question answering dataset with tables from Wikipedia, paired with corresponding questions, answers, and supporting table cells.

We further evaluate the methods on three summarization datasets. **XSum** (Narayan et al., 2018) contains BBC articles from 2010 to 2017, along with their summaries, each consisting of one highly abstractive sentence. **SAMSum** (Gliwa et al., 2019) is a dataset of messenger conversations about daily-life topics, annotated with short summaries. **PubMed** (Tang et al., 2023) is a collection of medical scientific articles where the goal is to summarize the conclusions of the authors based on the description of a medical experiment.

Although our primary focus is on domain-specific tasks, Appendix A.7 shows the results of applying SCOPE to a generalist model fine-tuned with instructions on the Alpaca dataset (Taori et al., 2023).

## 4.2    METRICS

We present in what follows the different metrics used for each task. Having in mind the limitations of BLEU and ROUGE metrics (resp. used for data-to-text generation and summarization as standard metrics for each task) and regarding our research objectives, we focus on faithfulness metrics that evaluate the generation with respect to the input context.

**BLEU (Data-to-text).**    Traditional metric to assess the similarity between the generated text and given gold references. In the context of data-to-text generation, it has shown limitations especially when reference text diverges from the input data (Dhingra et al., 2019).

**PARENT Recall (PAR, Data-to-text).**    (Dhingra et al., 2019) A standard n-gram based faithfulness proxy metric for data-to-text introduced to address the limitations of BLEU. It assesses how well the candidate text replicates relevant entities from the data by measuring its n-gram recall against entities in the structured input. Unlike BLEU, PARENT Recall directly compares to the structured input, making it a more suitable measure of faithfulness.

**NLI Score (NLI. Data-to-text).**    Proposed by Dušek & Kasner (2020), this metric adapts NLI models to data-to-text. It first computes the entailment probabilities of atomic input facts extracted from the structured data by the candidate text, characterizing *omissions*. A second score measures *hallucinations* by computing the entailment probability of the generated text by the sum of all the facts in the input data. The resulting NLI score is the minimum of all the entailment probabilities, assessing the overall faithfulness of the generated text.

**ROUGE-L (R-L, Summarization).**    (Lin, 2004) Traditional n-gram overlap summarization metrics between the generated and the gold reference. Similarly to BLEU, it has known limitations (Fabbri et al., 2021) regarding faithfulness evaluation.

**AlignScore (AL, Summarization).**    (Zha et al., 2023) A recent state-of-the-art entailment metrics. It measures the information alignment between the summary and the source article on a 0-1 scale, using a RoBERTa model (Liu et al., 2019) trained on a unified set of entailment tasks.

**QuestEval (Summarization).**    (Scialom et al., 2021) A reference-free evaluation metric for summarization. It assesses the semantic alignment between the source article and the generated summary by generating and answering questions about their content. QuestEval uses a question generation and answering pipeline, leveraging a pre-trained language model, to compute a similarity score between the information in the source and the summary.

**FactCC (Summarization).**    (Kryscinski et al., 2020) A factual consistency metric for summarization. It evaluates the factual alignment between the summary and the source article on a binary scale. FactCC relies on a fine-tuned BERT model, trained specifically to detect factual consistency through synthetic data generated by introducing factual errors into summaries.

**GPT-4 preference (Both tasks).**    Previous works (Gilardi et al., 2023; Chiang & Lee, 2023) have shown that powerful LLMs, like GPT-4 can serve as effective proxies for human evaluation. To provide a scalable human-like assessment of the generations' faithfulness, we use GPT-4 for pairwise preference evaluation. Given a an input and two texts, the model is asked which sample is more

faithful to the input data. This metric yields win, loss, and tie rates against the standard fine-tuning baseline. Details regarding GPT-4 preference evaluation can be found in Appendix A.8.

### 4.3 MODELS AND BASELINES

We experiment on LLAMA2-7B (Touvron et al., 2023) and MISTRAL-7B (Jiang et al., 2023), two recent LLMs. For all baselines, hyperparameters were carefully determined from a grid-search following recommendations in reference articles, using NLI Score for data-to-text generation and AlignScore for summarization as objectives. All training details and hyperparameters can be found in Appendix A.

**Supervised fine-tuning (SFT).**   This is the standard fine-tuning approach where the pre-trained model $p_{\text{LM}}$ is optimized using MLE on the *full* training dataset $\mathcal{D}$. We train for 3 epochs and choose the model according to NLI Score for data-to-text generation and AlignScore for summarization.

**PMI decoding (PMI).**   (van der Poel et al., 2022) PMI reduces hallucinations by penalizing "ungrounded tokens" when next-token entropy is high, adjusting probabilities using a context-less model with hyperparameters $\lambda$ and $\tau$.

**Context-aware decoding (CAD).**   (Shi et al., 2023) Similar to PMI, CAD downweights probabilities using a context-less model, with an adjustment factor controlled by $\alpha$.

**Critic-driven decoding (CRITIC).**   (Lango & Dusek, 2023) CRITIC improves generation by using, for each dataset, a model trained to differentiate context-supported tokens. It factors the model probability and generates samples based on a score combining the token probability and the classifier's context likelihood, adjusted by $\lambda$.

**CLIFF.**   (Cao & Wang, 2021) CLIFF is a training method that leverages a contrastive learning framework, where more positive samples are generated through a back-translation method, while negative samples are created using Named Entity Recognition (NER) models and different mask-and-generate methods. We choose the MASKREL baseline, which demonstrate strong overall results in the original paper. Initially designed for encoder-decoder models, we reimplemented the method for decoder-only architectures.

**SCOPE (ours).**   Models trained following SCOPE framework. For the experiments, we tune the noise level $\alpha$ by selecting the value that yields the highest NLI Score or AlignScore on the validation set. As detailed in Section 6, we restrict our search of $\alpha$ to the $[0.4, 0.6]$ interval, which corresponds to a zone where the BLEU/ROUGE scores does not decrease significantly. The selected value for each dataset and model can be found in Table 10.

CLIFF and SCOPE are methods that present a training method, while CAD and PMI modify the decoding process. CRITIC trains a model to modify the decoding process. We highlight that all baselines have been trained on the same amount of annotated samples, since decoding methods are applied to a fully fine-tuned model.

## 5 RESULTS

We now present the results of SCOPE and of the baselines on the data-to-text generation and text summarization tasks introduced above.

**SCOPE improves faithfulness over all tasks and domains.**   According to automatic faithfulness metrics, training with SCOPE gives consistent and significant improvements in faithfulness compared to standard fine-tuning, as presented in Tables 2 and 3. For data-to-text generation (Table 2), training models with SCOPE show significant improvements over standard fine-tuning, with an increase of up to 8.2 and 5.5 points PARENT and NLI Score respectively. For text summarization (Table 3), SCOPE demonstrates an increase of up to 8.8 points in AlignScore. On most datasets, SCOPE scores slightly lower BLEU and ROUGE scores than other baselines, especially on the abstractive XSum dataset. Previous work (Tanya Goyal, 2022) highlighted the saturation of summarization benchmarks and the limitations of reference-based metrics like BLEU and ROUGE in evaluating the summarization capabilities of recent LLMs. Given the high faithfulness scores achieved by SCOPE on both tasks, we suggest that this decrease in BLEU and ROUGE may indicate SCOPE's tendency to deviate from standard fine-tuning and to disfavor irrelevant generations.

| | ToTTo | | | E2E | | | FeTaQA | | | WebNLG | | |
|---|---|---|---|---|---|---|---|---|---|---|---|---|
| | NLI | PAR | BLEU | NLI | PAR | BLEU | NLI | PAR | BLEU | NLI | PAR | BLEU |
| LLAMA2-7B | | | | | | | | | | | | |
| SFT | 46.42 | 80.55 | - | 92.62 | 86.41 | 41.81 | 39.06 | 78.68 | 39.72 | 79.36 | 79.19 | 48.37 |
| CAD | 46.33 | 80.59 | - | 92.74 | 86.35 | 41.32 | 39.67 | 78.93 | 39.64 | 79.62 | 79.45 | **48.95** |
| CRITIC | 46.22 | 80.66 | - | 92.70 | 86.45 | 41.82 | 39.10 | 78.67 | 39.94 | 79.47 | 79.51 | 48.83 |
| PMI | 46.36 | 80.51 | - | 92.66 | 86.42 | 41.78 | 39.23 | 78.52 | 39.71 | 79.54 | 79.30 | 48.45 |
| CLIFF | 46.69 | 80.77 | - | 92.64 | 86.47 | 41.78 | 39.67 | 79.11 | **40.48** | 79.92 | 79.31 | 47.99 |
| SCOPE (ours) | 51.88* | 86.11* | - | 94.64* | 87.21* | 38.70 | 42.97* | 83.40* | 38.96 | 83.42* | 85.95* | 48.16 |
| LLAMA2-13B | | | | | | | | | | | | |
| SFT | 46.56 | 80.47 | - | 93.39 | 86.42 | 41.26 | 39.66 | 79.22 | 40.72 | 80.07 | 78.14 | 48.77 |
| CAD | 46.68 | 80.66 | - | 93.25 | 86.41 | 41.24 | 39.56 | 79.21 | 40.65 | 82.55 | 79.06 | **49.78** |
| CRITIC | 46.59 | 80.73 | - | **93.58** | 86.44 | 41.17 | 39.82 | 79.51 | 40.37 | 80.24 | 78.37 | 49.10 |
| PMI | 46.55 | 80.46 | - | 93.43 | 86.35 | 41.23 | 40.03 | 79.32 | 40.77 | 80.02 | 78.38 | 49.02 |
| CLIFF | 47.04 | 80.68 | - | 92.42 | 86.47 | **41.49** | 38.85 | 79.06 | **41.05** | 80.15 | 79.09 | 48.16 |
| SCOPE (ours) | 54.27* | 86.58* | - | 91.61 | 87.37* | 39.09 | 41.91 | 83.30* | 36.77 | 84.44* | 87.26* | 48.02 |
| MISTRAL-7B | | | | | | | | | | | | |
| SFT | 46.70 | 80.79 | - | 92.64 | 85.88 | 41.16 | 39.90 | 79.31 | 41.47 | 84.71 | 80.58 | 50.85 |
| CAD | 46.40 | 80.37 | - | 92.28 | 85.80 | 40.65 | 39.99 | 79.61 | 41.18 | 85.26 | 80.55 | 50.72 |
| CRITIC | 46.72 | 80.75 | - | 92.80 | 85.97 | 40.00 | 39.55 | 79.50 | 41.43 | 84.62 | **80.71** | 50.94 |
| PMI | 46.48 | 80.33 | - | 92.80 | 85.88 | **41.18** | 39.80 | 79.30 | 41.49 | 84.86 | 80.58 | 50.87 |
| CLIFF | 47.30 | 80.89 | - | 92.86 | 85.99 | 41.23 | 40.25 | 79.45 | **41.88** | 84.29 | 80.52 | 50.57 |
| SCOPE (ours) | 53.45* | 89.01* | - | 93.43 | 87.09* | 40.44 | 42.03 | 81.49* | 40.33 | 86.39* | 80.41 | 52.20 |

Table 2: Performance comparison on the test set of ToTTo, E2E, FeTaQA, and WebNLG. Note that the missing BLEU results are due to the absence of gold references in the test set of ToTTo. * denotes faithfulness scores statistically significantly higher than the SFT baseline.

| | SAMSum | | | | XSum | | | | PubMed | | | |
|---|---|---|---|---|---|---|---|---|---|---|---|---|
| | Align | FactCC | QEval | R-L | Align | FactCC | QEval | R-L | Align | FactCC | QEval | R-L |
| LLAMA2-7B | | | | | | | | | | | | |
| SFT | 80.66 | 78.51 | 44.83 | 45.20 | 56.25 | 74.63 | 31.99 | 34.92 | 46.89 | 35.84 | 34.60 | 24.58 |
| CAD | 81.65 | 79.37 | 45.01 | 45.01 | 57.58 | 77.83 | 32.26 | 33.73 | 52.68 | 43.05 | 33.65 | 22.50 |
| CRITIC | 81.52 | 77.66 | 45.18 | 44.81 | 55.80 | 74.23 | 32.03 | 34.15 | 48.02 | 37.56 | 33.71 | 23.80 |
| PMI | 81.03 | 77.29 | 44.95 | **45.15** | 56.29 | 74.33 | 31.99 | 34.90 | 48.03 | 36.34 | 34.45 | 23.56 |
| CLIFF | 81.30 | 76.68 | 44.77 | 44.72 | 57.46 | 74.70 | 32.23 | **35.58** | 45.64 | 37.56 | 34.06 | 23.97 |
| SCOPE | **83.67*** | 81.93 | **46.65*** | 42.15 | 65.10* | 89.05* | 38.76* | 27.58 | 58.17* | 58.63* | 38.53* | 24.00 |
| LLAMA2-13B | | | | | | | | | | | | |
| SFT | 81.59 | 78.63 | 44.10 | 44.60 | 56.53 | 75.75 | 31.72 | 36.14 | 47.51 | 38.93 | 34.83 | 24.02 |
| CAD | 81.35 | 80.59 | 44.21 | 43.43 | 57.22 | 77.45 | 31.99 | 35.89 | 52.81 | 47.79 | 34.67 | 23.17 |
| CRITIC | 81.14 | 78.14 | 44.40 | 42.88 | 56.53 | 75.16 | 31.81 | 35.97 | 49.06 | 40.46 | 34.63 | 22.35 |
| PMI | 81.82 | 78.14 | 44.04 | 44.75 | 56.56 | 75.47 | 31.75 | **36.20** | 50.87 | 36.79 | 34.82 | 23.32 |
| CLIFF | 81.61 | 76.80 | 44.96 | 44.19 | 56.52 | 75.27 | 31.67 | 36.10 | 45.60 | 40.76 | 34.30 | **24.39** |
| SCOPE | **84.20*** | 81.69 | **46.45*** | 44.98 | 66.03* | 84.06* | 37.17* | 31.59 | 58.68* | 61.22* | 39.10* | 23.85 |
| MISTRAL-7B | | | | | | | | | | | | |
| SFT | 82.59 | 75.75 | 31.25 | 44.20 | 57.20 | 75.76 | 31.25 | 36.25 | 43.60 | 35.10 | 33.32 | 25.07 |
| CAD | 83.10 | 79.37 | 45.52 | 43.98 | 57.31 | 78.55 | 31.32 | 35.24 | 45.36 | 42.75 | 31.72 | 23.63 |
| CRITIC | 82.76 | 79.24 | 45.63 | 44.07 | 57.65 | 74.67 | 31.81 | 33.68 | 46.80 | 38.78 | 33.13 | 23.55 |
| PMI | 82.45 | **80.46** | 45.49 | 44.17 | 57.47 | 76.76 | 30.83 | 36.17 | 44.08 | 37.86 | 32.59 | 24.37 |
| CLIFF | 82.50 | 79.24 | 45.60 | **44.30** | 58.20 | 75.33 | 31.83 | **37.14** | 45.90 | 40.61 | 34.18 | 25.50 |
| SCOPE | **83.70*** | 80.59 | **46.21*** | 42.72 | 62.17* | 84.36* | 36.33* | 24.61 | 55.37* | 48.55* | 37.01* | 24.03 |

Table 3: Performance comparison on the test set of SAMSum, XSum and PubMed. * denotes faithfulness scores statistically significantly higher than the SFT baseline.

**Baselines present mixed results on faithfulness metrics.** Summarization-focused baselines (CAD, PMI, CLIFF) show an overall increase in AlignScore on SAMSum, XSum and PubMed (Table 3). However, the improvements on XSum remain marginal compared to SCOPE's results. For data-to-text generation, all baselines show minimal to no faithfulness improvement over SFT (Table 2). Depending on the methods, we identified two reasons that could explain these mixed results. First, CLIFF, CRITIC, and PMI were originally designed for smaller encoder-decoder models. We suspect that differences in architecture and the number of parameters in larger, more recent LLMs may limit their effectiveness. Secondly, CAD, PMI, CLIFF were mainly designed for general summarization tasks, we suspect that for data-to-text generation, which require further adaptation, these methods may fall short.

**GPT4-as-a-judge evaluation.** To further assess their performances, all methods applied to LLAMA-2-7B were compared to standard fine-tuning, with GPT-4 used as the evaluator. Results are presented in Tables 4 and 5. Across all datasets, SCOPE consistently shows a much higher win rate than other methods, confirming its efficiency in improving faithfulness. For the baselines, especially in data-to-text generation tasks, we observe a noticeable high tie rate. This indicates that a significant proportion of the samples are considered equivalent in quality to the standard fine-tuning samples. Consequently, it suggests that these methods have not adequately addressed the faithfulness issues related to fine-tuning.

| | ToTTo | | | E2E | | | FeTaQA | | | WebNLG | | |
|---|---|---|---|---|---|---|---|---|---|---|---|---|
| | Win% | Tie% | Loss% | Win% | Tie% | Loss% | Win% | Tie% | Loss% | Win% | Tie% | Loss% |
| CAD | 3,47 | 93,11 | 3,42 | 1,79 | 92,20 | 6,01 | 7,59 | 86,78 | 5,62 | 8,70 | 82,1 | 9,20 |
| PMI | 2,82 | 94,33 | 2,85 | 0.49 | 99.02 | 0.49 | 5.90 | 86.01 | 8.10 | 7.98 | 84.26 | 7.76 |
| CRITIC | 4.37 | 91.5 | 4.13 | 0,87 | 98.00 | 1,14 | 5,85 | 89,49 | 4,67 | 6,90 | 86,25 | 6,85 |
| CLIFF | 14.57 | 72.37 | 13.06 | 3.14 | 92.15 | 4.71 | 20.92 | 58.66 | 20.42 | 14.90 | 67.96 | 17.14 |
| SCOPE (ours) | **35.03**[*] | 47.26 | 17.71 | **11.04**[*] | 84.79 | 4.17 | **29.96** | 45.53 | 24.51 | **29.85**[*] | 55.93 | 14.22 |

Table 4: GPT-4 preference results of CAD, PMI, CRITIC, CLIFF and SCOPE versus SFT with LLAMA-2-7B on ToTTo, E2E, FeTaQA and WebNLG. Results with [*] are statistically significantly higher than all other baselines.

| | SAMSum | | | XSum | | | PubMed | | |
|---|---|---|---|---|---|---|---|---|---|
| | Win% | Tie% | Loss% | Win% | Tie% | Loss% | Win% | Tie% | Loss% |
| CAD | 21.73 | 62.27 | 16.00 | 42.98 | 18.36 | 38.67 | 53.82 | 11.93 | 34.25 |
| PMI | 9.89 | 80.71 | 9.40 | 24.06 | 52.66 | 23.27 | 37.31 | 26.30 | 36.39 |
| CRITIC | 18.93 | 63.00 | 18.07 | 35.50 | 28.10 | 36.40 | 38.84 | 22.02 | 39.14 |
| CLIFF | 25.89 | 45.67 | 28.45 | 50.63 | 12.00 | 37.38 | 41.74 | 17.43 | 40.83 |
| SCOPE (ours) | **58.12**[*] | 8.42 | 33.46 | **61.03**[*] | 2.64 | 36.33 | **74.50**[*] | 22.75 | 2.75 |

Table 5: GPT-4 preference results of CAD, PMI, CRITIC, CLIFF and SCOPE versus SFT with LLAMA-2-7B on SAMSum, XSum and PubMed. Results with [*] are statistically significantly higher than all other baselines.

**Further validation through human evaluation.**
In addition to using automatic faithfulness metrics and GPT-4 preference judgments, we conduct human evaluations to comprehensively assess the quality of SCOPE generations. We distribute different sets of 25 ToTTo samples to 5 annotators, totaling 125 samples. Each sample includes a table, one generation from SCOPE and one from SFT, using LLAMA-2-7B. Annotators are tasked with rating which of the two

| | Win% | Tie% | Loss% |
|---|---|---|---|
| SFT | 15.2 | 44.8 | 40.0 |
| SCOPE | **40.0** | 44.8 | 15.2 |

Table 6: Human preference results of SCOPE versus SFT on ToTTo test set with LLAMA-2.

descriptions is more faithful to the table. They are asked to put the emphasis on *faithfulness exclusively*, meaning that although a generation may contain factually correct details, these additions are deemed less desirable than a generation that strictly relies on the information provided in the table. Full experimental details are described in Appendix E. The results are presented in Table 6. The descriptions generated by SCOPE are preferred twice as often as those by the associated SFT. The results closely match those obtained with GPT4-as-a-judge, further validating the soundness of our approach. We present some samples of SCOPE and SFT generations in Appendix D.

## 6    ANALYSIS OF SCOPE

In this section, we propose to analyze the effect of undesired responses generated by our unfaithful sampling method on the overall performances of SCOPE. By varying the value of $\alpha$ in the noisy data generation process (Algorithm 1), we can simulate different degrees of hallucinations due to the influence of $p_{LM}$. In this analysis, we examine the impact of negative samples on preference learning.

**How does the value of $\alpha$ affect the training dynamics?** The choice of $\alpha$ is critical. When $\alpha$ is low, the negative samples are too close to the model's own approximation of the underlying data distribution. During the preference-tuning stage, the model struggles to maximize the gap between the likelihood of the clean and noisy samples while maintaining the high likelihood of the clean ones. This causes the model to downweight the likelihood of both samples, leading to degeneracies, see Figure 2a. Conversely, when $\alpha$ is high, the generated noisy samples are barely grounded in the input context, making it easy to distinguish between $y$ and $y^-$ under $p_\theta(\cdot \mid c)$. In this case, the

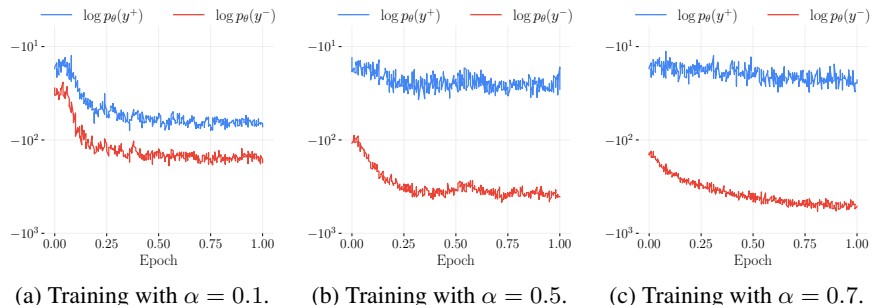

(a) Training with $\alpha = 0.1$.   (b) Training with $\alpha = 0.5$.   (c) Training with $\alpha = 0.7$.

Figure 2: Preference training dynamics with LLAMA-2-7B as noise level $\alpha$ increases on ToTTo dataset. Illustration of the three different regimes during preference training. Blue (resp. red) curve corresponds the log probability of the reference labels (resp. of the synthetic unfaithful samples).

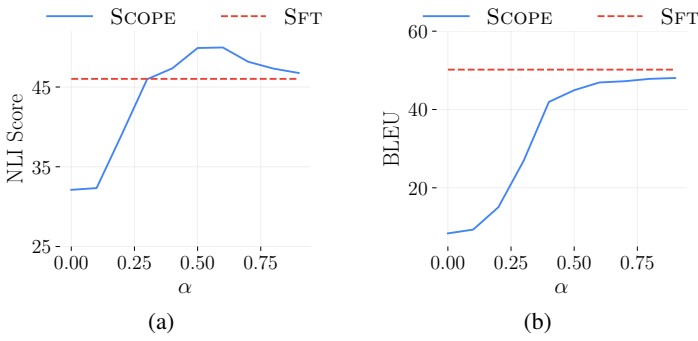

(a)                                                          (b)

Figure 3: Evolution of NLI Score and BLEU with $\alpha$ on ToTTo validation set with LLAMA-2-7B.

model learns very little compared to its fine-tuned counterpart, see Figure 2c. Therefore, $\alpha$ should be chosen to balance the noisy generation between being too similar to the reference texts and too easy to discriminate. This scenario is illustrated on Figure 2b for $\alpha = 0.5$. The likelihood of the references does not decrease, and the likelihood of the noisy samples diverges less abruptly than in Figure 2c, providing a more effective learning signal.

**How does the negative samples construction affect generation quality?**   For low values of $\alpha$, we observe noticeable degeneracies, evidenced by text repetitions. This is shown in Figure 3b, where BLEU scores decrease abruptly with lower values of $\alpha$. As dicussed in Section 5, in the $[0.4, 0.6]$ interval, the decrease in BLEU appears to be more closely related to the generated outputs diverging from standard fine-tuning patterns, rather than a noticeable decline in fluency. Regarding optimization efficiency, the three regimes observed in Figure 2 can also be identified in Figure 3a, that describes the evolution of the NLI score as a function of $\alpha$. Below a certain level of noise, degeneracies also impact the NLI score. Increasing $\alpha$ beyond a certain point yields no further improvement, as both BLEU and NLI scores converge to the results of standard fine-tuning. As a result, searching for $\alpha$ in the interval $[0.4, 0.6]$ seems to yield the best performances. We observe similar patterns in text summarization tasks, see Appendix B. A quantitative and qualitative analysis of the noisy samples can be found in Appendix C.

## 7   CONCLUSION

Faithfulness hallucinations are a common issue in standard fine-tuned LLMs, and existing methods developed to mitigate these hallucinations yield mixed results with recent LLM models. In contrast, we demonstrate that employing a two-stage method, distinct from standard fine-tuning, effectively addresses typical challenges. Our key contributions include the automatic and self-supervised construction of a preference dataset tailored for the model, along with a framework that enables preference learning. Notably, our approach, SCOPE, consistently enhances the faithfulness of generated responses across various data-to-text and summarization tasks, significantly outperforming existing solutions as assessed by relevant automatic faithfulness metrics, evaluations using GPT-4 and human judges. We provide an analysis of the main factors contributing to the successful deployment of this method, illustrating its performance quantitatively and qualitatively with typical samples.

## 8 LIMITATIONS

Although this work explores classical generation tasks, it is important to note that some of these tasks, particularly WebNLG and E2E, are of relatively limited complexity. Additionally, we limited our experiments to 7B models due to computational constraints. While this choice allows us to effectively demonstrate our approach, larger models could potentially yield different insights. Future work could validate the scalability and effectiveness of the proposed methods on larger model architectures. Lastly, this study primarily relied on automatic evaluation metrics. While these metrics have shown value, particularly in assessing faithfulness, their performance across the diverse domains of our datasets remains less explored. Ideally, a broader human evaluation would provide a more nuanced understanding of the results. However, given the resource and logistical constraints of conducting such evaluations, automatic metrics serve as a practical solution within the scope of this work, even if they have certain limitations.

## 9 ACKNOWLEDGMENTS

This work has been partly funded through project ACDC ANR-21-CE23-0007. This project was provided with computing AI and storage resources by GENCI at IDRIS thanks to the grants 20XX-AD011014053R2, 20XX-A0151014638, 20XX-A0171014638 and 20XX-A0151014627 on the supercomputer Jean Zay's V100/A100 partition.

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

# A EXPERIMENTS

**Implementation details.** Our code is based on Pytorch (Paszke et al., 2019) and Huggingface Wolf et al. (2020). Experiments were ran on NVidia 80GB A100 GPUs. BLEU is computed using the SacreBLEU (Post, 2018) implementation. For NLI metric we use the model available at `https://huggingface.co/cross-encoder/nli-deberta-v3-large`.

## A.1 BASELINES

For each baseline we choose the best hyper-parameters by conducting a grid-search. We initially conducted the search over ranges disclosed in original publications and refined based on our own experiments.

### A.1.1 CONTEXT-AWARE DECODING

Table 7 shows the best hyperparameters for CAD method. In the original paper, it is recommended to select $\alpha$ between 0 and 1, with 0.5 being a suitable choice.

| | ToTTo | FeTaQA | WebNLG | E2E | SAMSum | XSum | PubMed |
| --- | --- | --- | --- | --- | --- | --- | --- |
| | $\alpha$ | $\alpha$ | $\alpha$ | $\alpha$ | $\alpha$ | $\alpha$ | $\alpha$ |
| LLAMA-2-7B | 0.01 | 0.05 | 0.03 | 0.03 | 0.05 | 0.40 | 0.40 |
| LLAMA-13B | 0.01 | 0.01 | 0.07 | 0.01 | 0.3 | 0.10 | 0.40 |
| MISTRAL-7B | 0.01 | 0.09 | 0.01 | 0.04 | 0.04 | 0.30 | 0.20 |

Table 7: Best Context-Aware Decoding (CAD) $\alpha$ hyperparameter.

### A.1.2 POINTWISE MUTUAL INFORMATION

Table 8 shows best hyperparameters for PMI method.

| | ToTTo | FeTaQA | WebNLG | E2E | SAMSum | XSum | PubMed |
| --- | --- | --- | --- | --- | --- | --- | --- |
| | $(\lambda, \tau)$ | $(\lambda, \tau)$ | $(\lambda, \tau)$ | $(\lambda, \tau)$ | $(\lambda, \tau)$ | $(\lambda, \tau)$ | $(\lambda, \tau)$ |
| LLAMA-2-7B | (0.07, 3.25) | (0.07, 3.25) | (0.05, 3.5) | (0.06, 3.25) | (0.20, 3.25) | (0.15, 3.25) | (0.20, 3.25) |
| LLAMA-13B | (0.07, 3.25) | (0.05, 3.25) | (0.05, 3.25) | (0.05, 3.40) | (0.15, 3.25) | (0.10, 3.75) | (0.15, 3.25) |
| MISTRAL-7B | (0.06, 3.5) | (0.07, 3.25) | (0.05, 3.25) | (0.09, 3.25) | (0.05, 3.25) | (0.20, 3.25) | (0.15, 3.25) |

Table 8: Best PMI Decoding (PMI) $(\lambda, \tau)$ hyperparameters.

### A.1.3 CRITIC-DRIVEN DECODING

For the classifier, we replace the original XLM-RoBERTa-base (Conneau et al., 2019) with a stronger DebertaV3-large (He et al., 2023) model allowing for much larger contexts, since the linearized data did not fit in the context-window of XLM-RoBERTa-base. In our experiments, we trained a classifier on each dataset using the method *"base with full sentences"* reported to give the highest NLI score on WebNLG dataset in the original publication. Table 9 shows the best hyperparameters for the method.

| | ToTTo | FeTaQA | WebNLG | E2E | SAMSum | XSum | PubMed |
| --- | --- | --- | --- | --- | --- | --- | --- |
| | $\lambda$ | $\lambda$ | $\lambda$ | $\lambda$ | $\lambda$ | $\lambda$ | $\lambda$ |
| LLAMA-2-7B | 0.02 | 0.03 | 0.01 | 0.01 | 0.07 | 0.25 | 0.25 |
| LLAMA-13B | 0.03 | 0.07 | 0.01 | 0.06 | 0.25 | 0.10 | 0.75 |
| MISTRAL-7B | 0.05 | 0.01 | 0.01 | 0.10 | 0.05 | 0.75 | 0.50 |

Table 9: Best Critic-driven Decoding (CRITIC) $\lambda$ hyperparameter.

## A.2 HYPERPARAMETERS

**SCOPE $\alpha$.** Selected value of $\alpha$ for SCOPE for each dataset are presented in Table 10.

| | **ToTTo** | **FeTaQA** | **WebNLG** | **E2E** | **SAMSum** | **XSum** | **PubMed** |
|---|---|---|---|---|---|---|---|
| | $\alpha$ | $\alpha$ | $\alpha$ | $\alpha$ | $\alpha$ | $\alpha$ | $\alpha$ |
| LLAMA-2-7B | 0.5 | 0.5 | 0.5 | 0.6 | 0.5 | 0.5 | 0.4 |
| Llama-2-13B | 0.4 | 0.4 | 0.4 | 0.5 | 0.5 | 0.5 | 0.4 |
| MISTRAL-7B | 0.5 | 0.5 | 0.5 | 0.6 | 0.6 | 0.5 | 0.4 |

Table 10: Best SCOPE value of $\alpha$ for LLAMA-2-7B and MISTRAL-7B on ToTTo, FeTaQA, WebNLG, and E2E.

**Full SFT training.**

- LLAMA-2-7B. The SFT version of LLAMA-2-7B where fine-tuned using a batch size of 16, a learning rate of $2 \times 10^{-5}$, using a linear scheduler with a warm-up ratio of 0.1 on all datasets. The model is optimized with Adam optimizer.

- MISTRAL-7B. We used a batch size of 16, a learning rate of $2 \times 10^{-6}$ using a linear scheduler with a warm-up ratio of 0.1 on all datasets. The model is optimized with Adam optimizer.

**SCOPE training.**

- **Training $p_{\theta_0}$ on $\mathcal{D}_1$.** For training the fine-tuned version of each model on the split $\mathcal{D}_1$, we used the exact same setting than for the full SFT training described above, except that we only performed one epoch for LLAMA-2-7B and two epochs for MISTRAL-7B.

- **Preference tuning.** Regarding the hyperparameter of Equation (1), we set $\beta = 0.1$ for all models and datasets.

## A.3 FINE-TUNING ON HALF DATASETS

When fine-tuned on half the samples, we observe experimentally that the models have very close performances to the model fine-tuned on the full train set, see Tables 11 and 12. The models fine-tuned on half the samples are therefore a strong initialization for the subsequent stages of the method.

| | **WebNLG** | | **ToTTo** | | **E2E** | | **FetaQA** | |
|---|---|---|---|---|---|---|---|---|
| | **NLI** | **PARENT** | **NLI** | **PARENT** | **NLI** | **PARENT** | **NLI** | **PARENT** |
| LLAMA-2-7B | | | | | | | | |
| SFT on $\mathcal{D}_1$ | 86.6 | 72.0 | 45.6 | 80.4 | 80.9 | 82.2 | 36.2 | 76.6 |
| SFT on $\mathcal{D}$ | 87.4 | 82.1 | 46.0 | 80.2 | 87.4 | 86.9 | 37.5 | 77.1 |
| MISTRAL-7B | | | | | | | | |
| SFT on $\mathcal{D}_1$ | 87.2 | 81.6 | 46.5 | 80.3 | 87.0 | 87.4 | 34.1 | 74.6 |
| SFT on $\mathcal{D}$ | 87.5 | 81.9 | 46.7 | 80.1 | 86.5 | 85.2 | 34.1 | 74.8 |

Table 11: Results are on the validation sets. NLI Score and PARENT for models fine-tuned on half of the training set of a data-to-text datasets. On average, the score are slightly lower compared to models trained on the full dataset.

## A.4 ABLATION BY VARYING THE DATASET PROPORTIONS USED IN THE FIRST PHASE OF FINE-TUNING

Based on the observations in Appendix A.3, we chose to use 50% of the data for the first phase of fine-tuning given the considered datasets and tasks. Here, we present an ablation study on ToTTo. In this study, we fine-tuned a model on 25% (resp. 75%) of the dataset and preference-tuned on the remaining 75% (resp. 25%) with noisy samples. Results on the validation set, are shown in the table below. On automatic faithfulness metrics (NLI and PARENT), all splits yield comparable results, though a bit higher with a split of 50/50.

|  | XSum | | SAMSum | |
|---|---|---|---|---|
|  | AlignScore | Rouge-L | AlignScore | Rouge-L |
| LLAMA-2-7B | | | | |
| SFT on $\mathcal{D}_1$ | 56.2 | 33.8 | 80.5 | 43.2 |
| SFT on $\mathcal{D}$ | 56.4 | 35.2 | 82.6 | 45.2 |
| MISTRAL-7B | | | | |
| SFT on $\mathcal{D}_1$ | 57.3 | 35.1 | 81.9 | 44.7 |
| SFT on $\mathcal{D}$ | 57.3 | 36.2 | 82.5 | 45.2 |

Table 12: Results are on the validation sets. AlignScore and Rouge-L for models fine-tuned on half of the training set of a summarization datasets. Like for data-to-text generation, on average, the score are slightly lower compared to models trained on the full dataset.

| First phase trained on | NLI | PARENT |
|---|---|---|
| 25% | 49.57 | 86.08 |
| 50% | 50.64 | 86.34 |
| 75% | 49.07 | 84.10 |

Table 13: NLI and PARENT scores on the validation set of ToTTo when varying the proportion used in the first phase of fine-tuning and using the remaining split for the second phase of preference tuning.

## A.5 PREFERENCE LOSS

We chose to use DPO (Rafailov et al., 2023) for its seminal work and its widespread usage. But our self-supervised framework has no dependency with DPO and should also work with other preference tuning approaches. We tested with ORPO (Hong et al., 2024) and observed very similar results to DPO, see Table 14.

| Method | NLI | PARENT |
|---|---|---|
| SFT | 46.0 | 80.2 |
| SCOPE with DPO loss | 49.9 | 84.2 |
| SCOPE with ORPO loss | 49.3 | 85.9 |

Table 14: Results on the validation set of ToTTo with different preference optimization losses applied to LLAMA-2-7B.

## A.6 ABLATION ON THE VALUE OF $\beta$ IN PREFERENCE-TUNING STAGE

Table 15 presents faithfulness metrics as we change the value of $\beta$ in the preference-tuning phase of SCOPE. In the original DPO paper (Rafailov et al., 2023), authors use a value $\beta = 0.1$ which we found to also work well for SCOPE.

|  | ToTTo | | XSum | |
|---|---|---|---|---|
| $\beta$ | PARENT | NLI | ROUGE-L | AlignScore |
| 0.05 | 83.54 | 48.31 | 29.51 | 65.16 |
| 0.1 | **85.39** | **49.21** | 30.66 | **65.37** |
| 1 | 81.98 | 46.24 | 33.80 | 59.30 |
| 5 | 81.04 | 45.80 | **33.84** | 57.45 |

Table 15: The effect of different $\beta$ values on performance for ToTTo and XSum tasks.

## A.7 SCOPE ON INSTRUCTION-TUNED MODELS

We intentionally focused on a task-specific setup, targeting use cases where specialized models are most applicable. However, to explore SCOPE's performance in a general-purpose context, we

conducted additional experiments. Specifically, we fine-tuned a Llama-2-7b model on the Alpaca instruction dataset and compared it to a model fine-tuned using the SCOPE pipeline. Both models were evaluated on our initial tasks, including data-to-text and summarization. As shown in Table 16, SCOPE continues to demonstrate consistent gains in faithfulness according to our metrics. However, these improvements are smaller than those observed for domain-specific models, suggesting that SCOPE is particularly effective in specialized contexts.

| | ToTTo | | XSum | |
|---|---|---|---|---|
| | NLI | PARENT | AlignScore | Rouge-L |
| MODEL | | | | |
| SFT | 35.89 | 66.97 | 84.70 | **19.46** |
| SCOPE | **37.81** | **68.69** | **86.59** | 16.97 |

Table 16: On context-intensive tasks, SCOPE applied to generalist instruction-tuned models improves the faithfulness of the generation.

To ensure that these gains in faithfulness do not compromise reasoning capabilities, we benchmarked both models on tasks from the OpenLLM Leaderboard. The results indicate similar overall performance for both models, with SCOPE outperforming supervised fine-tuning (SFT) on tasks such as TruthfulQA, WinoGrande, and HellaSwag-tasks that require strong context comprehension rather than general knowledge. These results, presented in the appendix, reinforce our contributions. Nonetheless, a more comprehensive exploration of SCOPE's advantages in broader setups is left for future work.

| | ARC | HellaSwag | MMLU | TruthfulQA | Winogrande | Avg |
|---|---|---|---|---|---|---|
| SFT | **47.61** | 56.50 | **41.06** | 30.72 | 70.96 | 49.37 |
| SCOPE | 47.27 | **57.07** | 39.75 | **31.95** | **71.98** | **49.60** |

Table 17: Performance comparison between SFT and SCOPE on various tasks. Scores are percentages, and the best result for each task is highlighted in bold. Metrics: Accuracy is used for ARC, HellaSwag, MMLU, and Winogrande, while BLEU-Acc is used for TruthfulQA to evaluate faithfulness to the reference responses.

## A.8 GPT-4 PREFERENCE EVALUATION

As a proxy to a complete human evaluation, we conduct a GPT-4 preference evaluation comparing various methods to the SFT model. We ask the model to choose between two generations based on their faithfulness to the input data. We make sure to mitigate any position bias by randomly swapping the generations to be compared. We use the model `gpt-4-32k-0613` through the OpenAI API `https://platform.openai.com/docs/overview`. We use the following prompt for ToTTo, E2E and WebNLG:

> "You are a judge in a data-to-text competition. Your task is to determine which description more accurately reflects the information in a given data, ensuring that every detail in the text can be directly inferred from the data without adding any external information.
>
> Here is a data about **{Entity}**: **{Data}**
>
> Here are two descriptions of the data:
>
> Generation A: **{Generation A}**
>
> Generation B: **{Generation B}**
>
> Evaluate which description is more faithful to the data. Faithfulness means that every piece of information in the description must be directly inferable from the data and the description must not contain any additional information. Provide your answer in the following JSON format: {{"preferred_text": "<letter>"}} where

<letter> is "A" if Generation A is more faithful, "B" if Generation B is more faithful and "Tie" if both are equally faithful. "

for FeTaQA:

"You are a judge in a data question answering competition. Given a data and a question, your task is to determine which answer more accurately and faithfully responds to the question based on the information provided in the data, ensuring that every detail in the answer can be directly inferred from the data without adding any additional information.

Here is a data about **{Entity}**: **{Data}**

Given the data and the following question: **{Question}**

Here are two answers:

Answer A: **{Generation A}**

Answer B: **{Generation B}**

Evaluate which answer is more faithful to the data. Faithfulness means that every piece of information in the answer must be directly inferable from the data and the answer must not contain any additional information. Provide your answer in the following JSON format: {{"preferred_text": "<letter>"}} where <letter> is "A" if Answer A is more faithful, "B" if Answer B is more faithful and "Tie" if both are equally faithful. "

for XSum:

"You are a judge in an article summarization competition. Your task is to determine which summary more accurately and faithfully reflects the information in a given article, ensuring that every detail in the summary can be directly inferred from the article without adding any external information.

Here is an article: **{Article}**

Here are two summaries of the article:

Answer A: **{Summary A}**

Answer B: **{Summary B}**

Evaluate which summary is more faithful to the article. Faithfulness means that every piece of information in the summary must be directly inferable from the article and the summary must not contain any additional information. Provide your answer in the following JSON format: {{"preferred_text": "<letter>"}} where <letter> is "A" if Summary A is more faithful, "B" if Summary B is more faithful and "Tie" if both are equally faithful. "

for SAMsum:

"You are a judge in a messenger conversation summarization competition. Your task is to determine which summary more accurately and faithfully reflects the information in a given conversation, ensuring that every detail in the summary can be directly inferred from the conversation without adding any external information.

Here is a conversation: **{Article}**

Here are two summaries of the conversation:

Answer A: **{Summary A}**

Answer B: **{Summary B}**

Evaluate which summary is more faithful to the conversation. Faithfulness means that every piece of information in the summary must be directly inferable from the conversation and the summary must not contain any additional information. Provide your answer in the following JSON format: {{"preferred_text": "<letter>"}} where

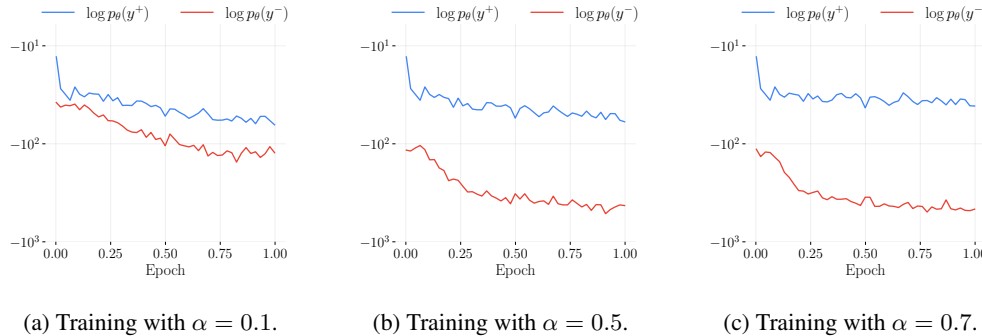

(a) Training with $\alpha = 0.1$.  (b) Training with $\alpha = 0.5$.  (c) Training with $\alpha = 0.7$.

Figure 4: Preference training dynamics with LLAMA-2-7B as noise level $\alpha$ increases on SAMSum dataset. We observe the same three different regimes during preference training than for data-to-text generation.

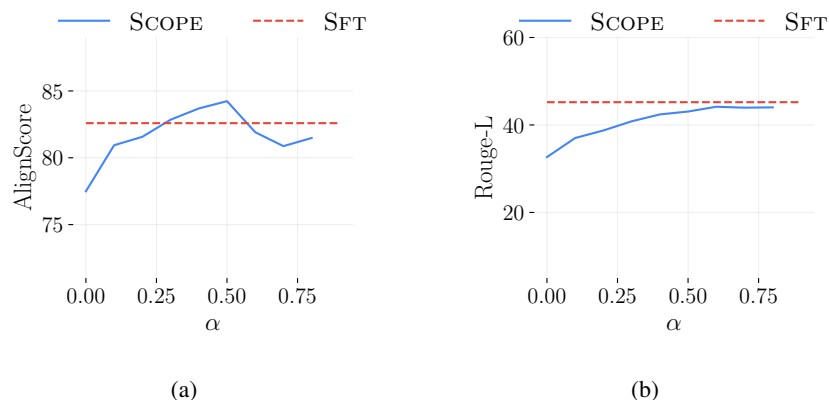

(a)  (b)

Figure 5: Evolution of AlignScore and Rouge-L with $\alpha$ on SAMSum validation set with LLAMA-2-7B.

> <letter> is "A" if Summary A is more faithful, "B" if Summary B is more faithful and "Tie" if both are equally faithful. "

## B  SCOPE ANALYSIS

We report here additional results supporting our analysis of SCOPE method of Section 6. The training dynamics of SCOPE on a summarization dataset is displayed on Figure 4 and the evolution of AlignScore and Rouge-L metrics on Figure 5. Overall, we observe similar patterns than for data-to-text generation.

## C  QUANTITATIVE AND QUALITATIVE ANALYSIS OF THE NOISY GENERATED SAMPLES

**Quantitative evaluation.**  To validate the effect of the noisy decoding process described in Algorithm 1, we plotted the evolution of PARENT and AlignScore as $\alpha$ increases on Figure 6.

**Qualitative assessment.**  Inspired by the error taxonomy presented in (Thomson & Reiter, 2020), we propose to annotate using three categories:
- **Incorrect**: statement that contradicts the data, includes incorrect number (including spelling out numbers as well as digits), incorrect named entity (people, places, organisations, etc) or other incorrect words. This corresponds to intrinsic errors.
- **Not checkable**: statement in the text that cannot be checked given the data. This corresponds to extrinsic information.

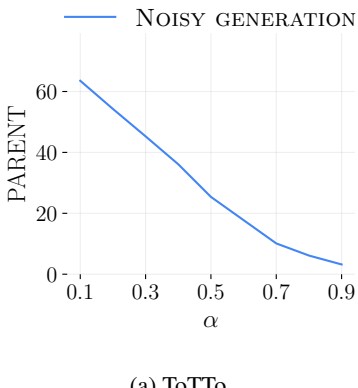
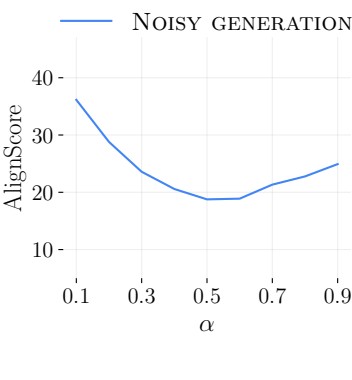

(a) ToTTo.    (b) XSum.

Figure 6: Evolution of the faithfulness of noisy samples as the noise parameter $\alpha$ in the decoding process increases, evaluated using PARENT and AlignScore on ToTTo and XSum. For XSum, AlignScore initially decreases with increasing $\alpha$, followed by a slight uptick. We attribute this counterintuitive behavior to a limitation of AlignScore, which has not been tested with completely irrelevant data. Rather than approaching zero as expected, the score stabilizes at a constant nonzero value.

- **Other type of error**: statement that is irrelevant to the data.
As an ilustration, given the following data:
**Page Title:** List of Governors of South Carolina
**Section Title:** Governors under the Constitution of 1868
**Table:**

| # | Governor | Took Office |
|---|---|---|
| 74 | - | - |
| 75 | - | - |
| 76 | **Daniel Henry Chamberlain** | **December 1, 1874** |

Please refer to Table 18 for an overview of the noisy samples as $\alpha$ increases from 0.0 to 0.9.

## D    SAMPLES OF SCOPE AGAINST SFT

Tables 22 to 27 present qualitative winning examples of our model versus the fine-tuned model, judged by GPT-4. We additionally highlighted differences between both predictions, which further underscores the liability of GPT-4 as a judge for faithfulness.

Qualitative analysis on XSum reveals that the SFT baseline often struggles to ground its summaries in the provided article. In contrast, SCOPE produces fewer hallucinations but tends to directly quote portions of the article. For data-to-text tasks, the SFT baseline frequently infers extra information, whereas SCOPE remains closely aligned with the structured data.

## E    HUMAN EVALUATION PROTOCOL

For this study, we recruited five European annotators, all fluent in English, on a voluntary basis. For each sample, they were presented with an input table and two predictions from the LLAMA-2-7B model, trained using SCOPE and SFT, respectively. These predictions were randomly labeled as 'Text A' and 'Text B'. The models corresponding to A and B were randomly selected for each sample to prevent any positional bias. The annotators were instructed to choose between the options 'Text A is more faithful' or 'Text B is more faithful' depending on their preference for description A or B, respectively. If both texts are deemed equally faithful, the annotators should select 'Tie'. If both

| $\alpha$ | Noisy generation |
|---|---|
| 0.0 | Daniel Henry Chamberlain was the 76th governor of South Carolina in 1874. |
| 0.1 | Daniel Henry Chamberlain was the 76th Governor of South Carolina and served from 1874. He was the first governor elected by popular vote. |
| 0.2 | Daniel Henry Chamberlain was the 19th and final Governor of South Carolina, serving from 1874 until 1876. |
| 0.3 | Daniel P. Chamberlain was elected as governor in 1854. |
| 0.4 | In 1876, the first woman elected as governor in the United States was Daniel Henry Chamberlain. |
| 0.5 | Daniel Henry Chamberlain, Jr. served as a U.S. Representative and served as the 7th Governor of South Carolina from December 18, 1974. He was a member of the Democratic Party. |
| 0.6 | Tags: Daniel Henry Chamberlain was born in 1887, and died on December 1, 1962. He was the son of Daniel Henry Chamberlain, who served as a politician and lawyer in South Carolina. |
| 0.7 | Danielle Hatcher Chamberlain served as a U.S. Senator from 1843-1847 and was elected as a Governor of Mississippi in 1847. She was elected again for another term in 1870. |
| 0.8 | Oshima-yukihisa-kōki was discovered by Japanese amateur astronomer Atsushi Sugiyama on October 25, 1995 at the Okayama Astrophysical Observatory. |
| 0.9 | Heteromastix piceaformis piceaformis (B) species group (Heteromastix) complex (B). |

Table 18: At low levels of noise, the noisy sample is close to the supervised fine-tuned model, being overall faithful to the context while adding unsupported information (extrinsic error). As $\alpha$ increases, the influence of the unconditional model causes the sample to increasingly contradict the context (intrinsic error), eventually making it entirely irrelevant.

descriptions have one or several faithfulness issues, they should both be considered unfaithful and rated as 'Tie'. The following instructions were provided to the annotators:

**Instructions for Faithfulness Evaluation**

Your task is to assess which text description is more faithful to the corresponding table. In this context, a text is considered **faithful** if all information it contains is directly supported by the content of the table.

- If the description introduces any unsupported or incorrect information, it should be rated as **unfaithful**.

- If both descriptions contain one or more faithfulness issues, rate them as a **Tie**.

To guide your evaluation:

- Carefully compare each detail in the description with the table to ensure accuracy.

- A description should not distort, omit, or add information that is not present in the table.

- If you notice even a single instance of unsupported information in a description, it should be rated as unfaithful.

- If both descriptions have one or several faithfulness issues, they should both be considered unfaithful and rated as 'Tie'.

Please choose between the following options for each comparison:

- **Text A is more faithful**

- **Text B is more faithful**

- **Tie** (if both descriptions are equally faithful or contain faithfulness issues)

# F ON THE SIGNIFICANCE IMPROVEMENTS OF SCOPE AGAINST THE OTHER BASELINES

**Faithfulness metrics.** We performed independent two-sample t-tests to assess whether there were statistically significant differences in the mean values of specified metrics between the baseline SFT and comparison model SCOPE. This test was chosen as it accounts for unequal variances and assumes independence between the two samples. For each metric, we calculated the t-statistic and corresponding p-value, allowing us to evaluate the likelihood that observed differences in means arose by chance. The results provide a statistical basis for determining the significance of observed variations across datasets. Using a standard p-value of 0.05, SCOPE is statistically significantly better than SFT across the vast majority of datasets, metrics and models.

| | ToTTo | | FeTaQA | | WebNLG | | E2E | |
|---|---|---|---|---|---|---|---|---|
| | PARENT | NLI | PARENT | NLI | PARENT | NLI | PARENT | NLI |
| LLAMA2-7B | 3.19e−50 | 4.73e−17 | 2.21e−12 | 3.68e−3 | 1.11e−31 | 4.55e−4 | 7.18e−3 | 4.01e−3 |
| LLAMA2-13B | 4.91e−60 | 6.22e−31 | 1.37e−9 | 9.26e−2 | 1.06e−55 | 1.85e−4 | 1.48e−3 | 1.02e−2 |
| MISTRAL-7B | 1.86e−103 | 2.26e−24 | 1.08e−3 | 1.13e−1 | 7.64e−1 | 1.68e−1 | 4.51e−5 | 2.57e−1 |

Table 19: p-values of paired t-tests between SCOPE and SFT for data-to-text datasets.

| | SAMSum | | | XSum | | | PubMed | | |
|---|---|---|---|---|---|---|---|---|---|
| **Model** | Align | FactCC | QEval | Align | FactCC | QEval | Align | FactCC | QEval |
| LLAMA2-7B | 1.25e−3 | 1.1e−1 | 4.26e−2 | 3.56e−80 | 3.54e−69 | 4.67e−144 | 3.29e−11 | 4.79e−21 | 3.47e−8 |
| LLAMA2-13B | 1.48e−2 | 0.1216 | 3.7e−3 | 1.10e−69 | 3.16e−55 | 5.36e−160 | 1.06e−9 | 3.27e−16 | 2.53e−7 |
| MISTRAL-7B | 3.98e−3 | 3.56e−2 | 4.38e−1 | 5.37e−73 | 3.16e−55 | 3.33e−189 | 1.20e−17 | 3.05e−22 | 1.10e−12 |

Table 20: p-values of paired t-tests between SCOPE and SFT for summarization datasets.

**Pairwise rating.** To assess whether our SCOPE improves significantly over the other baselines based on our GPT-4 win-tie-lose pairwise preference evaluations, we perform the McNemar's statistical test to determine if the observed difference in wins is likely due to chance or if it reflects a truly performance difference.

- **Null hypothesis**: There is no significant difference in performance between SCOPE and given baseline. Any difference in win counts is due to random chance.

- **Alternative hypothesis**: SCOPE performs significantly better than the considered baseline.

To do this, we count the number of samples SCOPE wins over SFT while the compared baseline loses to it ($N_{AB}$) and vice versa ($N_{BA}$) without taking into account the ties. The McNemar's test formula is given by:

$$\chi^2 = \frac{(N_{AB} - N_{BA})^2}{N_{AB} + N_{BA}}$$

Under the null hypothesis, $\chi^2$ follows a chi-square distribution with 1 degree of freedom.

We consider a standard p-value of 0.05. A p-value less than 0.05 means we reject the null hypothesis. Here are the p-values on the GPT-4-as-a-judge evaluations:

| **Comparison** | **Totto** | **WebNLG** | **FeTaQA** | **E2E** | **SamSum** | **XSum** | **PubMed** |
|---|---|---|---|---|---|---|---|
| SCOPE vs SFT | 3.696e−97 | 3.127e−23 | 9.7e−4 | 3.559e−14 | 1.171e−25 | 6.744e−153 | 3.944e−41 |
| SCOPE vs PMI | 9.492e−7 | 7.7e−3 | 6.744e−1 | 4.78e−2 | 1.3e−3 | 6.269e−55 | 2.305e−19 |
| SCOPE vs CRITIC | 1.473e−8 | 1.95e−2 | 2.1e−1 | 2.7e−3 | 5.41e−11 | 4.781e−74 | 2.313e−4 |
| SCOPE vs CAD | 1.226e−7 | 6.792e−5 | 9.39e−2 | 1.33e−2 | 1.23e−7 | 2.611e−59 | 1.522e−11 |
| SCOPE vs CLIFF | 1.226e−11 | 3.745e−11 | 6.25e−2 | 5.6e−4 | 2.04e−6 | 3.025e−4 | 1.314e−21 |

Table 21: p-values of the McNemar's test on GPT-4 evaluation results

The results from McNemar's test show that:

(i) SCOPE shows consistently a significant improvement over the SFT baseline.

(ii) Most of the comparisons between SCOPE and the other baselines are statistically significant (p-value < 0.05) on ToTTo, WebNLG, E2E, SamSum, and XSum with the exception of FeTaQA.

| Input | SFT | SCOPE |
|---|---|---|
| Denmark international Fischer, 21, has been with the Amsterdam club since 2011 and scored 24 goals in 79 league games. The forward visited Boro, who were promoted to the Premier League this season, for a medical and agreed terms. ""It's the greatest league in the world and I feel really ready,"" Fischer told Danish TV station TV2. Eight-times capped Fischer has agreed a three-year deal. Fischer emerged as a key player at Ajax under recently departed coach Frank De Boer, alongside compatriots Lasse Schone and Christian Eriksen - the latter whom he will play against in next season's top flight when Boro face Tottenham Hotspur. | **ROUGE-L**: 11.43 / **AL**: 0.4 Middlesbrough have signed Ajax striker Nicklas Bendtner for an undisclosed fee. | **ROUGE-L**: 23.52 / **AL**: 82.65 Boro have signed Ajax forward Nicky Fischer for an undisclosed fee, with the striker targeting "the greatest league in the world" in next season's Premier League. |
| The inquiry, carried out by Professor Sir Bruce Keogh, England's NHS medical director, was set up following the Mid-Staffordshire Hospital scandal. A report into high death rates at two hospitals in Mid-Staffordshire earlier this year found there were hundreds more deaths than expected, with patients let down by poor standards of care. The table below shows how the 14 NHS trusts covered by the review scored in two measures of mortality. A score much above 100 is higher than the national average. All of the data has been standardised to take into account the different case-mix at hospitals - for example, hospitals that treat more elderly patients might be expected to have higher death rates. | **ROUGE-L**: 32.00 / **AL**: 20.67 A review of NHS trusts in England has found that 11 of 14 trusts have been rated as performing above the national average for the number of patients who died. | **ROUGE-L**: 17.77 / **AL**: 90.87 NHS trusts covered by an inquiry into high mortality rates at hospitals in Staffordshire scored above the national average in two measures, BBC data shows. |
| The Robins have lost 11 of their past 12 league games, winning just once in that time, and are 20th in the table - two points above the relegation zone. ""It is a horrendous statistic,"" Johnson told BBC Radio Bristol. ""I'm ashamed of that. It hurts. We have to keep fighting."" Former Barnsley boss Johnson arrived at Ashton Gate in February 2016 and steered Bristol City clear of relegation last season. A promising start this term saw his side occupy fifth spot in October, but Saturday's loss at the City Ground saw them drop below Forest - who had not won since 2 December. ""It was a frustrating watch for all of us,"" Johnson said. ""We were good in the second half and had a couple of chances we needed to take. ""The Championship was never going to be easy - we are in a rebuilding phase. ""We spoke at the start of the season about history makers and it was supposed to be positive - not bad history. ""We've got to make sure we make history in positive ways and we can still do that this year. We need to fight our way back up the table. I believe that we can."" | **ROUGE-L**: 42.10 / **AL**: 4.0 Bristol City manager Lee Johnson says he is "ashamed" of his side's poor form after they lost 2-0 at Nottingham Forest. | **ROUGE-L**: 21.87 **AL**: 86.5 Bristol City boss Lee Johnson "hurts" to be "ashamed" of his side's "horrendous" recent Championship run and insists they can still fight their way back up the table. |

Table 22: XSum random winning samples. For the sake of clarity, we purposely choose articles of reasonable size. Red highlights facts that are hallucinations. Yellow highlights facts that are more faithful to the input.

| Input | SFT | SCOPE |
|---|---|---|
| Joana: wana play some games?
Sandy: what games?
Joana: maybe dixit?
Joana: or carcassonne?
Sandy: why not?
Joana: ok, I'll bring them 2 u
Sandy: great :) | Joana will bring Dixit and Carcassonne to Sandy. | Joana will bring Dixit and Carcassonne ==games== for Sandy. |
| Ana: You sleeping?
Catherine: Not yet.
Ana: Wanna go visit grandma tomorrow? I miss her.
Catherine: Yeah that would be nice :) I'll call you when I wake up
Ana: Oki :) sleep well, good night.
Catherine: Good night, u too. | Ana and Catherine will visit grandma tomorrow. | Catherine will ==call Ana when she wakes up== tomorrow to visit grandma tomorrow. |
| "Ellie: hey, are you at the university?
Aaliyah: yep
Camille: I'm sick :/
Ellie: shiet....that's bad
Ellie: Aali, where are we having our classes now?
Aaliyah: 342 on the second floor" | ==Ellie, Aaliyah and Camille are at the university==. They have their classes in 342 on the second floor. | Aaliyah is at the university in 342 on the second floor. ==Camille is sick==. |

Table 23: SAMsum random winning samples. ==Red== highlights facts that are hallucinations. ==Yellow== highlights facts that are more faithful to the input.

| Input | SFT | SCOPE |
|---|---|---|
| **1992 Cork Senior Hurling Championship**
Player: Brian Corcoran
Club: Erin's Own
Tally: 0-44 | Erin's Own's Brian Corcoran was the championship's top scorer with 0-44. | Erin's Own Brian Corcoran was the ==1992 Cork Senior Hurling Championship=='s top scorer with 0-44. |
| **2015–16 Reading F.C. season**
Matches: 7 November 2015
Matches: Cardiff City
Matches: 2–0
Matches: Reading
Matches: Cardiff | On 7 November, Reading lost 2–0 away to Cardiff City. | Reading lost 2–0 to Cardiff City on ==7 November 2015==. |
| **United National Party**
Candidate: Ranil Wickremesinghe
%: 48.43% | Ranil Wickremesinghe ==won== the ==presidential election== with 48.43% of the vote. | Ranil Wickremesinghe received 48.43% of the vote. |
| **2018 Monster Energy NASCAR Cup Series**
Race: Big Machine Vodka 400 at the Brickyard
Most laps led: Clint Bowyer Denny Hamlin
Winning driver: Brad Keselowski
Manufacturer: Ford | In the Big Machine Vodka 400 at the Brickyard, Clint Bowyer and Denny Hamlin led the laps for Brad Keselowski in Ford. | Brad Keselowski drove Ford to win the Big Machine Vodka 400 at the Brickyard, Clint Bowyer and Denny Hamlin ==led the most laps==. |

Table 24: ToTTo random winning samples. ==Red== highlights facts that are hallucinations. ==Yellow== highlights facts that are more faithful to the input.

| Input | SFT | SCOPE |
|---|---|---|
| **Ciudad Ayala** population metro: 1777539 leader title: "City Manager" type: City population density: 1604.0 government type: Council-manager government utc offset: -6 country: Mexico | The city of Ayala, Mexico is led by the City Manager and has a population density of 1604.0. The UTC offset for the city is -6 and it has a population of 1777539. | The city of Ciudad Ayala, in Mexico, has a UTC offset of -6. The city is led by the City Manager and has a population density of 1604.0 and a population of 1777539. ==The government type is the council-manager government==. |
| **The Velvet Underground** associated band/associated musical artist: Theatre of Eternal Music | The Velvet Underground is associated with the Theatre of Eternal Music. | The Velvet Underground is associated with the ==band== Theatre of Eternal Music. |
| **Expect a Miracle** artist: Brian Kelly (composer & pianist) type: Compilation Album runtime: 54.56 preceded by: Pools of Light (Brian Kelly album) | Brian Kelly is a pianist and composer who created ==the Expect a Miracle==. It is 54.56 minutes long and was preceded by Pools of Light. | Expect a Miracle is a ==compilation album== written by Brian Kelly (composer & pianist) and preceded by Pools of Light. It has a runtime of 54.56. |
| **Nord (Year of No Light album)** release date: 2006-09-06 genre: Sludge metal genre: Post-metal | ==The Year of No Light==, a post-metal and sludge metal album, was released on 6th September 2006. | The ==genre== of ==Nord (Year of No Light album)== is sludge metal and post-metal and it was released on 6th September 2006. |
| **Liselotte Grschebina** nationality: Israel | The nationality of the ==artist==, Liselotte Grschebina, is Israeli. | The nationality of Liselotte Grschebina is Israel. |

Table 25: WebNLG random winning samples. ==Red== highlights facts that are hallucinations. ==Yellow== highlights facts that are more faithful to the input.

| Input | SFT | SCOPE |
|---|---|---|
| **The Vaults**
name: The Vaults
priceRange: £20-25
near: Raja Indian Cuisine | The Vaults is a moderately priced place to eat near Raja Indian Cuisine. | The Vaults is near Raja Indian Cuisine and has a price range of £20-25. |
| **The Cricketers**
name: The Cricketers
eatType: restaurant
food: Chinese
priceRange: cheap
customer rating: 5 out of 5
area: riverside
familyFriendly: yes
near: All Bar One | The Cricketers is a Chinese restaurant in the riverside area near All Bar One. It is family friendly and has a high price range and a low customer rating. | The Cricketers is a Chinese restaurant near All Bar One in riverside. It has a cheap price range and a customer rating of 5 out of 5 and is family friendly. |
| **The Phoenix**
name: The Phoenix
eatType: restaurant
priceRange: less than £20
area: city centre
area: riverside
familyFriendly: yes
near: Raja Indian Cuisine | The Phoenix is a family friendly restaurant located near Raja Indian Cuisine in the city centre. It is in the low price range and is located near the river. | The Phoenix is a family friendly restaurant located near Raja Indian Cuisine in the city centre area by the riverside with a price range of less than £20. |
| **Blue Spice**
name: Blue Spice
eatType: restaurant
area: riverside
familyFriendly: no
near: Rainbow Vegetarian Café | Blue Spice is a restaurant located near Rainbow Vegetarian Café in the riverside area. It is not family-friendly and has a price range of less than £20. | Blue Spice is a non family-friendly restaurant located near Rainbow Vegetarian Café in riverside. |
| **The Mill**
name: The Mill
eatType: pub
eatType: restaurant
food: Fast food
area: riverside
familyFriendly: yes
near: Café Sicilia | The Mill is a fast food restaurant in the riverside area near Café Sicilia. It is a pub that is kid friendly and has a price range of more than £30. | The Mill is a family friendly pub style fast food restaurant near Café Sicilia in riverside. |

Table 26: E2E random winning samples. Red highlights facts that are hallucinations. Yellow highlights facts that are more faithful to the input.

| Input | Question | SFT | SCOPE |
|---|---|---|---|
| Mauricio Ortega (discus thrower)
Position: 1st
Event: Discus throw
Competition: Bolivarian Games
Notes: 59.67 m
Year: 2013 | What ratio of total votes did Pierre Bertho earn during the 1946 Dahomey and Togo by-election? | Pierre Bertho of the Popular Republican Movement won with 644 of the 818 votes. | Pierre Bertho of the Popular Republican Movement earned 644 of the 818 total votes. |
| Candidate: Doug Forrester
Candidate: Jon Corzine
%: 53.5
%: 43.0 | What were the results of Forrester and Jon Corzine? | Forrester lost to Jon Corzine by 43.0% to 53.5%. | Forrester received 43.0% of the vote, while Jon Corzine received 53.5% of the vote. |
| Model(s): Biturbo i 2500
Displacement: 2.8 L (2,790 cc)
Model(s): 228, 430, 222 E, Spyder 2800, Karif
Displacement: 2.5 L (2,491 cc) | What was the difference in engine displacement of the 222 E model compared to the Biturbo? | The 228 E model has a 2.8 L (2,790 cc) engine, larger than the 2.5 L (2,491 cc) Biturbo i 2500. | The 222 E model had an engine displacement of 2.8 L (2,790 cc) compared to the Biturbo's 2.5 L (2,491 cc). |
| Title: Skyfall
Title: Spectre
Year: 2015
Role: Eve Moneypenny
Year: 2012 | In what films did Harris play Eve Moneypenny? | Harris played Eve Moneypenny in the 2012 film Skyfall and its 2015 sequel Spectre. | Harris played Eve Moneypenny in Skyfall (2012) and Spectre (2015). |
| rank: 6
name: Pihlajavesi
area (km²): 712.59 | How large is the area of the Pihlajavesi lake? | The Pihlajavesi lake has an area of 712.59 square kilometres, which is the sixth largest lake in Finland. | The Pihlajavesi lake has an area of 712.59 km². |

Table 27: FeTaQA random winning samples. Red highlights facts that are hallucinations. Yellow highlights facts that are more faithful to the input.

