# OpenReview forum: "SCOPE: A Self-supervised Framework for Improving Faithfulness in Conditional Text Generation"
_ICLR.cc/2025/Conference — ICLR 2025 Poster_

### Official Review · Reviewer_aJYu · 2024-11-01

**Soundness:** 3
**Presentation:** 3
**Contribution:** 3
**Rating:** 6
**Confidence:** 4

**Summary:**

The paper presents SCOPE, a self-supervised framework that enhances the faithfulness of LLMs in conditional text generation tasks like summarization and data-to-text generation. The framework employs a two-stage approach: initial fine-tuning followed by preference tuning using contrastive learning. The key innovation lies in its method of automatically generating synthetic unfaithful samples by blending grounded and context-free generation, which are then used to train the model to prefer context-aligned outputs over hallucinated ones. Through comprehensive evaluation across multiple datasets, SCOPE demonstrates significant improvements in output faithfulness, achieving up to 14% gains according to automatic metrics, with additional improvements confirmed through both GPT-4 and human evaluations.

**Strengths:**

- An innovative self-supervised approach to improving LLM faithfulness through synthetic unfaithful sample generation
- The method shows strong performance across multiple benchmarks, with notable improvement in data-to-text tasks.
- The paper explains the methodology and experimental setup well, using helpful figures and clear explanations of core ideas, such as the noisy sample generation and preference tuning.

**Weaknesses:**

- While SCOPE performs well on general summarization and data-to-text tasks, evaluating it on domain-specific datasets (e.g., biomedical or financial) could better demonstrate its robustness in high-stakes settings where hallucinations are especially problematic.
- SCOPE is tested on 7B models only; applying it to models of different sizes would clarify its scalability and help determine whether it generalizes effectively to larger different models. Perhaps the improvement using SCOPE on larger models will be marginal.
- The evaluation on summarization task used ROUGE and AlignScore (an entailment metric). While SCOPE scores slightly lower ROUGE scores than other baselines, the authors did not use other faithfulness metrics in summarization research to further evaluate.

**Questions:**

1. What is the rationale behind splitting the dataset into 50-50 for the two phases? Would other split also work?
2. Do you anticipate that SCOPE would perform similarly on domain-specific datasets, like those in healthcare or finance?
3. Can SCOPE’s approach to unfaithful sample generation and preference tuning generalize to larger or different model architectures?
4. The $\alpha$ for CAD is set to very small values for tasks other than XSum. Why is it the case?
5. The main evaluation metrics used are BLEU/NLI/ROUGE/AL. Did you consider any additional faithfulness metrics or error types that might capture different facets of faithfulness?

---

> ### Author Response · Authors · 2024-11-24
>
> Thank for your review and feedback. You will find below our answers to your questions. We have added new experimental results following each of your suggestions.
>
> **W1 & Q2. Evaluation on domain-specific datasets**
>
> Thanks for the suggestion. Benchmarking SCOPE on a domain where hallucinations are problematic could better demonstrate the robustness of the approach.
>
> We followed the experimental setup of the recent article [1] on medical evidence summarization, applied to a subset of the PubMed dataset [2]. In this setting, we have evaluated SCOPE and the baselines on this additional dataset (see Tables 3 and 5 in the paper). SCOPE’s training method demonstrates strong faithfulness gains, compared to baselines and vanilla fine-tuning. We updated the paper to include this new dataset, hoping this will strengthen the robustness of the approach.
>
> **W2 & Q3. SCOPE on larger models**
>
> We followed your suggestion (shared with reviewer eSUQ), we extended our experiments to include SCOPE and the baseline methods on a larger Llama-2-13B model. The results are conclusive regarding scalability: SCOPE achieves a significant improvement in faithfulness compared to the baseline, demonstrating its consistent effectiveness across model scales. We have included these findings in the paper (see Table 3), as they further validate the robustness and scalability of the method.
>
> **W3 & Q5. Using other metrics for summarization**
>
> We also followed your suggestion and re-evaluated the models using two additional widely used metrics for assessing faithfulness in text summarization:
>
> - **QuestEval** [3], a QA-based evaluation method.
> - **FactCC** [4], a model trained to identify conflicts between the summary and the source document.
>
> The results, now included in the paper (see Table 3), show that models trained with SCOPE achieve substantial improvements on these metrics compared to the baselines. We believe these findings further reinforce the reported effectiveness of SCOPE.
>
> **Q1. Rationale behind splitting the datasets**
>
> Our initial idea was to modify the training process by leveraging the training samples differently while maintaining the same amount of annotated data. We developed the SCOPE pipeline and tested it using a default 50/50 split, which we retained for the following experiment reasons:
>
> - Fine-tuning on the first half of the dataset yields performance comparable to fine-tuning on the full training set (see Appendix A.3).
> - Across all datasets, SCOPE with a 50/50 split consistently delivers strong results.
>
> To further justify this split, we conducted an ablation study on ToTTo. In this study, we fine-tuned a model on 25% (resp. 75%) of the dataset and preference-tuned on the remaining 75% (resp. 25%) with noisy samples. Results on the validation set are shown in the table below. On automatic faithfulness metrics (NLI and PARENT), all splits yield comparable results, though a bit higher with a split of 50/50. Based on these results, we recommend sticking with a base value of 50/50 when using SCOPE. We have updated the paper to include this experiment in Appendix A.4.
>
> | First phase trained on | NLI | PARENT |
> | --- | --- | --- |
> | 25% | 49.57 | 86.08 |
> | 50% | 50.64 | 86.34 |
> | 75% | 49.07 | 84.10 |
>
> [1] Tang, L. *et al.* Evaluating large language models on medical evidence summarization. *npj Digit. Med.* **6**, 158 (2023). https://doi.org/10.1038/s41746-023-00896-7
>
> [2] Cohan, A. et al. (2018). A discourse-aware attention model for abstractive summarization of long documents. *Proceedings of the 2018 Conference of the North American Chapter of the Association for Computational Linguistics: Human Language Technologies, Volume 2 (Short Papers)*. Association for Computational Linguistics. https://doi.org/10.18653/v1/n18-2097
>
> [3] Thomas Scialom et al 2021. [QuestEval: Summarization Asks for Fact-based Evaluation](https://aclanthology.org/2021.emnlp-main.529). In *Proceedings of the 2021 Conference on Empirical Methods in Natural Language Processing*, pages 6594–6604, Online and Punta Cana, Dominican Republic. Association for Computational Linguistics.
>
> [4] Wojciech Kryscinski et al . 2020. [Evaluating the Factual Consistency of Abstractive Text Summarization](https://aclanthology.org/2020.emnlp-main.750). In *Proceedings of the 2020 Conference on Empirical Methods in Natural Language Processing (EMNLP)*, pages 9332–9346, Online. Association for Computational Linguistics.

---

> > ### Comment · Reviewer_aJYu · 2024-11-25
> >
> > Many thanks for the response and running the additional experiments in such a short period. After reading the updated paper and reviews from other reviewers, I decide to maintain my overall rating.
> >
> > In line 220, the hyperref is missing.

---

> > > ### Author Response · Authors · 2024-11-25
> > >
> > > We are pleased to hear that the new results meet your approval. Thank you for pointing out the missing hyperref. We have updated the paper accordingly.

---

### Official Review · Reviewer_eSUQ · 2024-11-02

**Soundness:** 2
**Presentation:** 2
**Contribution:** 3
**Rating:** 6
**Confidence:** 4

**Summary:**

The paper investigates the common issue of hallucinations in large language models (LLMs) during conditional text generation. To enhance faithfulness, the authors introduce SCOPE, a novel self-supervised fine-tuning approach that generates unfaithful samples and then trains the model to prefer context-grounded outputs over these fabricated examples. By using this approach, SCOPE achieves substantial improvements in generating grounded and faithful responses, as validated by automatic metrics and evaluations from both LLMs and human judges. The study's findings demonstrate SCOPE’s effectiveness across diverse tasks, including data-to-text and summarization, where it consistently outperforms existing self-supervised techniques.

**Strengths:**

1. The research problem addressed in this paper is significant, as faithfulness remains a major challenge for contextual generation in LLMs.
2. The authors propose an innovative method for generating negative examples for DPO training, utilizing two models—the fine-tuned model and the pre-trained model—selecting the next token based on a weighted combination of their distributions. This approach is independent of external tools like NER.
3. The experimental results demonstrate that their method outperforms baseline models across multiple tasks.

**Weaknesses:**

1. Some of the baselines appear to be training-free decoding methods, while others require training, similar to the method proposed in the paper. The authors should clarify which methods are training-free and which are not, as they require different resources to implement.
2. Additional ablation studies are needed, such as applying the same training framework with other negative sampling methods like NER replacement or negation. Otherwise, the effectiveness of the negative sampling itself remains uncertain.
3. Experiments should include the faithfulness score of the generated negative examples on the training set to directly demonstrate the impact of negative sampling.

**Questions:**

1. In the noisy generation step, is fluency in the output ensured? Since each token is selected based on the weighted sum of the original and fine-tuned LM distributions, could it happen that the token with the highest probability from the combined distribution is not among the top tokens in either individual distribution? For instance, if the two models’ distributions differ significantly, could an unsuitable token receive a relatively high score in each distribution and then emerge as the highest-scoring token after the weighted sum?
2. Is the generated negative examples all unfaithful? it would be useful to provide statistics or evaluation results indicating the proportion of genuine negative samples. This could potentially be estimated using the evaluation metrics applied in the experiments.

---

> ### Author Response · Authors · 2024-11-24
>
> Thanks for your review and relevant feedback. Please find below our answers to your concerns and questions:
>
> **W1. Presentation of the baselines**
>
> We updated the paper to better present the baselines. CAD/PMI are decoding methods while Critic and CLIFF require training. We hope this enhances the clarity of the paper.
>
> **W2. Ablation on negative sampling creation**
>
> We followed your suggestion and applied our same training framework using as an alternative to our method a NER replacement method: the noisy samples are generated by running a NER tool and replacing entities by other random entities of the same type. We experimented on ToTTo and XSum (results on the table below). This method however, did not report gains as high than SCOPE. We believe this experiment strengthens our initial intuitions regarding the importance of leveraging hallucination patterns that align closely with the fine-tuned model’s behaviors.
>
> | ToTTo |  |  |  | GPT-4 eval |  |
> | --- | --- | --- | --- | --- | --- |
> |  | PARENT | NLI | Win rate vs SFT | Tie rate vs SFT | Lose rate vs SFT |
> | SFT | 80.55 | 46.42 | 0% | 100% | 0% |
> | NER | 83.14 | 47.18 | 24.68% | 51.88% | 23.44% |
> | SCOPE | **86.11** | **51.88** | **35.03%** | 47.26% | 17.71% |
>
> | XSum |  |  |  | GPT-4 eval |  |
> | --- | --- | --- | --- | --- | --- |
> |  | ROUGE-L | AlignScore | Win rate vs SFT | Tie rate vs SFT | Lose rate vs SFT |
> | SFT | **34.92** | 56.25 | 0% | 100% | 0% |
> | NER | 27.58 | 46.09 | 27.34% | 5.14% | 67.52% |
> | SCOPE | 24.85 | **65.10** | **61.03%** | 2.64% | 36.33% |
>
> **W3-Q2. Adding metrics regarding negative examples**
>
> Following your suggestion, we scored the negative examples using our faithfulness metrics (PARENT for data-to-text and AlignScore for summarization). We updated the paper to include these analysis in Appendix C. Overall, we observe a clear decrease of faithfulness in the generated samples. Combined with a qualitative (human) analysis of the samples directly presented (see Table 18 in Appendix C), we believe we can conclude that negative sampling significantly introduces unfaithfulness issues in the generations.
>
> **Q1. Fluency of noisy generation step**
>
> You raise an important point: when combining the log probabilities of the fine-tuned model $p_\theta(y_t \mid y_{<t}, c)$ and the pre-trained model $p_{LM}(y_t \mid y_{<t})$, it is possible for a token with relatively low likelihood in both distributions to rank highly after the weighted combination. This could indeed lead to selecting less suitable tokens.
>
> This concern directly motivated our proposed noisy generation method. As described in Section 3.2 and Algorithm 1, we actually do not use a weighted mixture of distributions. Instead, at each step, we sample exclusively from either the fine-tuned or the base model ($\alpha_t$ is either 0 or 1). This ensures tokens are drawn from fluent distributions, avoiding the issue you describe.

---

> > ### Comment · Reviewer_eSUQ · 2024-11-25
> >
> > Thank you for your response, I have updated the score.

---

### Official Review · Reviewer_NUJj · 2024-11-07

**Soundness:** 3
**Presentation:** 3
**Contribution:** 3
**Rating:** 6
**Confidence:** 4

**Summary:**

This paper introduces a new learning framework for conditional text generation for the purpose of increasing the faithfulness of generated text with respect to the context input. The learning framework consists of two main steps: finetuning and contrastive learning. For contrasting learning, dispreferred samples are created in an unsupervised manner by utilizing a decoding method for generating certain tokens from the untrained language model, thereby introducing tokens which are not contextually consistent with the task. In several experiments, SCOPE is shown to be more effective than other baselines, including various decoding methods. SCOPE shows to be more faithful than traditional finetuning and SCOPE output is more preferred by humans and by GPT-4 judges.

**Strengths:**

Relatively novel methodology that uses the process of collaborative decoding for the purpose of generating dispreferred text. The method of collaborative decoding has been used in the past to combine text from a general LLM and an expert LLM to improve performance. In this setting, however, collaborative decoding is utilized to generate inconsistencies in the text randomly based on a threshold alpha.

Experiments in the tasks of summarization and data-to-text in several datasets which highlight the strength of SCOPE compared to traditional finetuning and other baseline decoding methods. Additionally, the large variety of metrics, human evaluation, and GPT-4 evaluation indicates that SCOPE is effective at increasing the faithfulness of the model for these two specific tasks.

**Weaknesses:**

Lack of human analysis of the unsupervised generated data: it is unclear whether the the dispreferred samples are worse due to the changes in the language distribution, or if the dispreferred samples contain information or text that is inconsistent with the context. If the unwanted generation mainly comes from the changes in the language distribution, then SCOPE is not learning to be more faithful to the context. This is a key distinction which may limit the interpretability of SCOPE for being a more faithful method of conditional text generation. One way to mitigate this concern would be to conduct a human analysis on a sample of generated data and determine if the generated data contains text inconsistent with the context or if the inserted text from the base LM is merely different in language distribution.

Lack of comparisons to instruction-tuned models. The methodology of SCOPE, at its essence, is very similar to the standard method of reinforcement learning with human feedback, where the model is first finetuned and then preference optimized with binary data consisting of a preferred sample and an unwanted sample. Due to the similarities, it would be valuable to compare SCOPE to RLHF methods and compare the degree of faithfulness of the model. There is evidence that instruction-tuned models are more "helpful" to users' inputs in the prompt, which correlates highly with faithfulness. Using the instruction-tuned model as a baseline, or running SCOPE on an instruction-tuned model is important to validate SCOPE as a novel, effective method.

**Questions:**

What are the impacts of training set size on the SCOPE methodology? It seems that for a small dataset, utilizing half of the dataset for finetuning and utilizing the other half for preference optimization may not be as effective as traditional finetuning.

Is there a way to rate the quality of the dispreferred samples in 3.2? For example, are there generated texts with more inconsistencies that are worse than other samples? Is there an impact on the potential difference in quality of the generated texts?

---

> ### Author Response · Authors · 2024-11-24
>
> **W1. Lack of human analysis and quality of dispreferred samples**
>
> We followed your suggestion and we conducted a human assessment of the negative samples. We summarize below the main findings, please refer to Appendix C for more details and to the examples in Table 18. These qualitative evaluations indicate that SCOPE dispreferred samples contains extrinsic and intrinsic errors, rather than exhibiting a shift in the language distribution.
>
> However, rating the quality of these samples systematically is not straightforward, and while this observation supports the importance of sample quality, our current findings remain qualitative. A more thorough study examining various quality dimensions of these samples could provide deeper insights into their role and impact, but we leave this as an avenue for future work.
>
> **W2. Comparison to instruction-tuned and RLHF baselines**
>
> Our method is indeed inspired by RLHF with human feedback but operates in a fully unsupervised setup. While our initial focus was on specialized models, we followed your suggestion and trained instruction-tuned models on the Alpaca dataset. One model was standardly tuned, while the other was trained using the SCOPE method, with noisy samples generated from the Alpaca dataset.
>
> We evaluated both models on our initial tasks (data-to-text and summarization), where SCOPE consistently demonstrated improvements in faithfulness (albeit with smaller gains compared to its performance on specialist models) (Appendix A.7, Table 16) and table below.
>
> |  | **ToTTo** |  | **XSum** |  |
> | --- | --- | --- | --- | --- |
> |  | **NLI** | **PARENT** | **AlignScore** | **Rouge-L** |
> | SFT | 35.89 | 66.97 | 84.70 | **19.46** |
> | SCOPE | **37.81** | 68.69 | **86.59** | 16.97 |
>
> Additionally, we verified that SCOPE does not impair reasoning capabilities, as evidenced by its performance on tasks from the OpenLLM benchmark (Appendix A.7, Table 17) and table below. We believe these results further underscore the robustness and versatility of the method.
>
> |  | **ARC** | **HellaSwag** | **MMLU** | **TruthfulQA** | **WinoGrande** | **Avg** |
> | --- | --- | --- | --- | --- | --- | --- |
> | SFT | **47.61** | 56.50 | **41.06** | 30.72 | 70.96 | 49.37 |
> | SCOPE | 47.27 | **57.07** | 39.75 | **31.95** | **71.98** | **49.60** |
>
> As a comparison to a RLHF baseline, we also report the performance of Llama-2-7b chat models on our tasks. While it performs better on summarization tasks (likely due to the prevalence and alignment of summarization with instruction-tuned models) it falls short compared to specialist models on data-to-text generation. This highlights the limitations of general-purpose models on highly specific tasks and underscores the continued need for task-specific models in certain use cases.
>
> |  | **ToTTo** |  |  | **E2E** |  |  | **FetaQA** |  |  |
> | --- | --- | --- | --- | --- | --- | --- | --- | --- | --- |
> |  | **NLI** | **PARENT** | **BLEU** | **NLI** | **PARENT** | **BLEU** | **NLI** | **PARENT** | **BLEU** |
> | CHAT | 48.19 | 75.73 | - | 73.73 | 74.69 | 10.62 | 34.05 | 74.26 | 24.48 |
> | SFT | 46.42 | 80.55 |  | 92.62 | 86.41 | **41.81** | 39.06 | 78.68 | **39.72** |
> | SCOPE | **51.88** | **86.11** | - | **94.64** | **87.21** | 38.70 | **42.97** | **83.40** | 38.96 |
>
> |  | **SamSum** |  | **XSum** |  |
> | --- | --- | --- | --- | --- |
> |  | **Rouge-L** | **AlignScore** | **Rouge-L** | **AlignScore** |
> | CHAT | 24.21 | **84.78** | 18.73 | **80.81** |
> | SFT | **45.20** | 80.66 | **34.92** | 56.25 |
> | SCOPE | 42.15 | 83.67 | 27.58 | 65.10 |
>
> **Q1. Impact of the training set size**
>
> Across all datasets whose sizes span from 10K (for FeTaQA) to 120K samples (for ToTTo), we found that SCOPE consistently improves over traditional fine-tuning (see Tables 2 and 3) for Llama2-(7b and 13b) and Mistral-7b.
>
> We then did not observe a direct correlation between training set size and the effectiveness of SCOPE. It is true that the largest datasets, ToTTo and XSum, show the highest gains in terms of automatic metrics. However, for the rebuttal, we also performed additional experiments on a selected subset of the PubMed dataset (following the experimental setup of [1]) which also demonstrates substantial improvement despite the dataset being relatively small (around 10K examples) (see Table 3 in the paper). Based on these observations, we believe SCOPE's effectiveness is more closely tied to the intrinsic difficulty of the task than to the size of the training set: XSum involves extreme summarization, ToTTo requires advanced table understanding, and PubMed focuses on summarizing technical medical articles.
>
> [1] Tang, L. *et al.* Evaluating large language models on medical evidence summarization. *npj Digit. Med.* **6**, 158 (2023). https://doi.org/10.1038/s41746-023-00896-7

---

> ### Author Response · Authors · 2024-11-24
>
> **Q2. Rate the quality of disprefered samples.**
>
> Thank you for your suggestion. One way to measure this effect is to explicitely generate samples with an increasing degree of noise. We have included both quantitative and qualitative analyses of the noisy samples in the appendix. Specifically, we plotted PARENT (for data-to-text generation) and AlignScore (for summarization) for the negative samples as a function of the noise parameter $\alpha$. As anticipated, both scores decrease as $\alpha$ increases. Qualitative observations indicate that this decline results from the introduction of extrinsic and intrinsic errors relative to the input context. For further details, please refer to Appendix C.

---

> ### Comment · Reviewer_NUJj · 2024-11-30
> **Scores**
>
> Thank you for responding, based on your response, I have changed my score of contribution to 3: good.

---

### Official Review · Reviewer_fCdZ · 2024-11-07

**Soundness:** 3
**Presentation:** 2
**Contribution:** 3
**Rating:** 6
**Confidence:** 2

**Summary:**

Scope is a self-supervised framework for improving the faithfulness of conditional text generation in LLMs. The approach uses a novel two-stage process: first fine-tuning on half the dataset, then using a mixture of that model and the base model to generate synthetic unfaithful samples for preference learning. The authors evaluate their method on data-to-text and summarization tasks.

**Strengths:**

- They study an important problem in the subfield -- factuality and faithfulness issue in conditional text generation. The research is motivated.
- Interesting two-stage training pipeline, and the approach to generate synthetic unfaithful samples.
- Provides analysis of preference training dynamics as well as the ablation for the noise parameter $\alpha$

**Weaknesses:**

- The preference learning stage uses β=0.1 for all models and datasets (A.2) without justification or ablation, despite this being a critical hyperparameter in preference learning.
- No evidence the method works on larger models. For their main claimed contribution ("improving faithfulness" in finetuning), testing on 7B models seems insufficient. Is it possible to include a few larger models, e.g. 14B, if not 70B scales.

**Questions:**

1. Have you investigated whether the method's effectiveness varies with model size? The limitation to 7B models leaves open questions about scalability.
2. The choice of α in [0.4, 0.6] seems empirically driven. Could it vary for different tasks / models?

---

> ### Author Response · Authors · 2024-11-24
>
> Thank you for your feedback and relevant suggestions. Please find below our answers to your concerns and questions:
>
> **W1. Ablation on $\beta$**
>
> We initially used a default value of 0.1 for $\beta$, a common choice in DPO experiments. We acknowledge the lack of ablation analysis for this parameter. To address this, following your suggestion, we conducted an ablation study (see the table below). Overall, we found that $\beta=0.1$ performs consistently well for both tasks. We have included this ablation in the paper.
>
> |  | ToTTo |  | XSum |  |
> | --- | --- | --- | --- | --- |
> | $\beta$ | PARENT | NLI | ROUGE-L | AlignScore |
> | 0.05 | 83.54 | 48.31 | 29.51 | 65.16 |
> | 0.1 | **85.39** | **49.21** | 30.66 | **65.37** |
> | 1 | 81.98 | 46.24 | 33.80 | 59.30 |
> | 5 | 81.04 | 45.80 | **33.84** | 57.45 |
>
> In the original DPO derivation, $\beta$ controls the strength of KL-regularization relative to the reference model, which in our case is the SFT model trained on half the dataset. As shown in the table, setting $\beta$ too low results in slightly worse performance, as the model may diverge excessively from the reference. Conversely, as $\beta$ increases, the gains diminish because the model is overly constrained and cannot sufficiently improve upon the reference SFT model.
>
> **Q1-W2. SCOPE on larger models**
>
> Indeed, our initial experiments focused only on 7B models, and we agree that evaluating SCOPE on larger models is indeed an insightful suggestion. In response, and in alignment also with feedback from reviewer aJYu, we conducted additional experiments with Llama2-13B on our tasks within the limited time available, and we updated the paper accordingly, see Tables 2 and 3. However, we were unable to extend our evaluation to 70B models due to computational constraints.
>
> For the 13B models, we observed very similar trends to those seen with 7B models, indicating that SCOPE demonstrates consistent effectiveness across different model scales.
>
> We provide below an excerpt of the results. More complete results are provided in Tables 2 and 3 in the paper.
>
> |  | ToTTo |  | XSum |  |
> | --- | --- | --- | --- | --- |
> |  | NLI | PARENT | ROUGE-L | AlignScore |
> | SFT | 46.56 | 80.47 | **36.14**  | 56.53 |
> | SCOPE | **54.27** | **86.58** | 31.59  | **66.03** |
>
> **Q2. Value of $\alpha$**
>
> We acknowledge that the choice of $\alpha$ is empirically driven. However, across all datasets and models, we consistently found that $\alpha$ values in the range [0.4, 0.6] yield effective results for SCOPE, with 0.5 often being the optimal choice (see Appendix A.2, Table 10). This consistency across tasks and models gives us confidence in recommending this range as a reliable starting point for most setups.

---

> ### Comment · Reviewer_fCdZ · 2024-12-03
> **Response to Authors**
>
> I thank the authors for explaining the parameter details.
>
> My original score still reflect my judgement about this paper. Nevertheless, I have improved my assessment regarding soundness and contribution.

---

### Official Review · Reviewer_hu8D · 2024-11-12

**Soundness:** 3
**Presentation:** 3
**Contribution:** 2
**Rating:** 6
**Confidence:** 4

**Summary:**

LLMs often struggle to generate fully grounded information to a provided context. There is a rich literature on the hallucination issues in generative models (https://arxiv.org/abs/2005.00661). This has become an even bigger priority as LLMs are being adapted widely.

This paper addresses the hallucination issues in LLMs and proposes a self-supervised method for synthetically generating a training set of negative responses. Using these negative examples, models are then trained to encourage the generation of grounded outputs over unfaithful ones using preference-based training.

The authors empirically demonstrate that the proposed method helps models generate more faithful texts, when evaluated automatically and by humans, on data-to-text generation and summarization.

**Strengths:**

The paper is trying to address generative hallucination, a common problem in generative models. The paper is fairly well written and easy to follow.

The paper brings new insights on the behavior of preference-tuning with the synthetically generated negative examples.

The proposed approach appears to improve response faithfulness, at a minor cost of fluency.

**Weaknesses:**

The method for synthetically generating negative examples is not novel, as the authors claim. Several papers (see some pointers below) in the past have used similar methods for generating negative or diverse sets of samples to better calibrate their models.

The paper should present a detailed analysis of the synthetic data being generated. For example, it would be nice to see faithfulness score distributions in negative examples. Also a qualitative analysis showing types of hallucinations being generated would have been very interesting.

Faithfulness is a pointwise metric, but the human evals mostly focus on SxS ratings. It would have been nice to see pointwise evaluations of faithfulness.

It would have been nice to see comparisons to other ways of generating negative examples, for example, prompting models to generate negative examples.

Other interesting work in this space that might be interesting to discuss here:

Brio or Slic: https://arxiv.org/abs/2203.16804, https://arxiv.org/abs/2210.00045
Calibrating models for faithfuness: https://arxiv.org/pdf/2310.08764
https://renzhaochun.github.io/assets/pdf/26596-Article%20Text-30659-1-2-20230626.pdf
https://arxiv.org/pdf/1910.08684 from the point of sub-sequence sampling
Comparison with CoT or planning based methods for controlling hallucinations.

-
After the rebuttals, I have adjusted the score. Thanks for conducting additional experiments and evaluations. They will be valuable to the readers.

**Questions:**

We are moving towards general purpose models, instead of a task-specific model. How does the proposed method generalize to other tasks, while achieving higher factuality/faithfulness for data-to-text generation and summarization? Will it be much harder to tune the value of alpha when tried with multiple tasks together?

SFT on a specific task often  suffers from catastrophic forgetting. It would be interesting to see how post training baselines do on these tasks. The post training often aims to improve faithfulness along other criteria.

Could you add a detailed analysis of synthetically generated data wrt faithfulness and fluency? Also how does the proposed method compare with other synthetic data generation methods?

Reference responses often have hallucinations. For example, XSum gold summaries often have hallucinated content. How does this effect the data generation? How do you ensure that an example is a positive example with respect to faithfulness?

---

> ### Author Response · Authors · 2024-11-24
>
> Thank you for sharing your feedback on our work; we truly appreciate it. Below, you'll find our responses to your concerns and questions:
>
> **W1. Novelty of the method**
>
> - We acknowledge that generating artificially bad samples is not a novel idea, and we appreciate the additional references you pointed out, which we will address in the related work. A key distinction between our method and some of the cited references is that many of them focus on improving post-fine-tuning generations by bootstrapping over model-generated sample pairs (e.g., BRIO, SLIC, SLIC-NLI). In contrast, our work introduces a novel fine-tuning approach that leverages training samples differently from standard fine-tuning, making SCOPE potentially complementary to these methods.
> - Another point we would like to highlight is the simplicity of our method. Implementing the noisy decoding process requires only a few lines of code, whereas existing training approaches often rely on significantly more complex setups involving additional models or tools. Our approach explicitly leverages the base pre-trained language model directly, avoiding external dependencies. This design aligns with the intuition that it introduces hallucination patterns more consistent with those the fine-tuned model, derived from the same pre-trained base, would naturally generate.
>
> **W2-Q3. Detailed analysis of the samples**
>
> We have revised the paper to include both quantitative and qualitative analyses of the noisy samples in the Appendix. Specifically, we plotted PARENT (for data-to-text generation) and AlignScore (for summarization) for the negative samples as a function of the noise parameter $\alpha$. As expected, both scores decline as $\alpha$ increases. Qualitative (human) observations indicate that this decline is caused by the introduction of extrinsic and intrinsic errors relative to the input context. Further details can be found in the updated Appendix C.
>
> **W3. Faithfulness pointwise evaluation**
>
> Thank you for the suggestion. We initially considered pairwise evaluation, considering that it was more suitable for human assessments. Following your suggestion, we also conducted a pointwise faithfulness evaluation of both SCOPE and SFT on XSum using GPT-4 (instead of human evaluation in the limited amount of time), utilizing the following prompt:
>
> >Your task is to determine whether the provided summary accurately reflects the information in a given article, ensuring that every detail in the summary can be directly inferred from the article without adding any external information. The summary can have one or more of the following errors:
> >
> >- **Extrinsic Information**: the summary contains new information not present in the source material.
> >- **Intrinsic Error**: the summary contradicts the source material.
> >
> >Rate the summary on a scale of 1 to 5 based on faithfulness:
> >
> >(5) **Completely faithful**: the summary is completely faithful to the article.
> >
> >(4) **Insignificant faithfulness errors**: the summary is mostly faithful, with slight inconsistencies not affecting main points.
> >
> >(3) **Partially faithful**: overall faithful, with a few inconsistencies with the article.
> >
> >(2) **Severe faithfulness errors**: nearly half of the summary is faithful, with severe deviation from main points.
> >
> >(1) **Completely unfaithful**: the entire summary is unfaithful to the article.
> >
> >First, identify the list of errors that the summary makes, ensuring each error is clearly stated and specific.
> >
> >Conclude the response with an evaluation using the following format: "Therefore, the score is: [insert score]."
>
> For data-to-text generation, we ask two human annotators to evaluate the faithfulness of 50 outputs from SCOPE and SFT using the same protocol as XSum described above.
>
> Please find below the average results (/5) of our assessments:
>
> |  | ToTTo | XSum |
> | --- | --- | --- |
> | SFT | 4 | 3.12 |
> | SCOPE | 4.1 | 3.6 |
>
> On XSum, both models exhibit faithfulness issues, but SCOPE shows a significant improvement over the SFT baseline. Overall, the pointwise evaluation confirms the trends found with pairwise assessment.
>
> **W4. Comparison with alternative noisy samples generation methods.**
>
> In response to similar feedback from other reviewers, we conducted an additional comparison using an alternative method for generating noisy samples, which involved randomly swapping entities from the reference samples. However, this approach did not yield gains as significant as those achieved with SCOPE. We believe this experiment reinforces our initial intuition about the importance of leveraging hallucination patterns that closely align with the behaviors of the fine-tuned model.

---

> > ### Author Response · Authors · 2024-11-24
> >
> > **Q1-Q2. SCOPE on general-purpose models**
> >
> > We initially focused on a task-specific setup, targeting use cases where specialized models are most applicable. However, we share your interest in assessing SCOPE's performance for generalist models. Following your suggestion, we fine-tuned a Llama-2-7b model on the Alpaca instruction dataset (SFT) and compared it to the same Llama-2-7b trained on Alpaca using the SCOPE pipeline, with noisy samples generated from the Alpaca dataset. Both models were then evaluated on our initial tasks (data-to-text and summarization). As shown in the table below, although not trained on ToTTo and XSum, SCOPE demonstrates consistent gains in faithfulness according to our metrics, though the improvements are smaller than those observed for domain-specific models, suggesting that SCOPE is particularly effective in specialized contexts.
> >
> > We also include the performance of the post-training method CAD applied to the generalist model fine-tuned on the Alpaca dataset. But the conclusions regarding the benefits of CAD are mixed: while we observe gains on ToTTo, on the summarization dataset XSum, it fails to improve the faithfulness of the SFT model. Tuning with SCOPE gives a consistent improvement.
> >
> > |  | **ToTTo** |  | **XSum** |  |
> > | --- | --- | --- | --- | --- |
> > |  | **NLI** | **PARENT** | **AlignScore** | **Rouge-L** |
> > | SFT | 35.89 | 66.97 | 84.70 | **19.46** |
> > | CAD | 37.03 | **70.60** | 84.43 | 17.78 |
> > | SCOPE | **37.81** | 68.69 | **86.59** | 16.97 |
> >
> > To ensure that the observed gains in faithfulness do not come at the cost of lowering reasoning capabilities, we benchmarked both models on tasks from the OpenLLM Leaderboard. The results indicate that both models perform similarly overall and SCOPE training does not degrade the performance of the generalist model. Notably, SCOPE outperforms SFT on tasks like TruthfulQA, WinoGrande, and HellaSwag-tasks that rely heavily on context comprehension rather than general knowledge. These results will be included in the appendix. While we believe this finding reinforces our contribution, a more comprehensive exploration of SCOPE's advantages in broader setups is left for future work.
> >
> > |  | **ARC** | **HellaSwag** | **MMLU** | **TruthfulQA** | **WinoGrande** | **Avg** |
> > | --- | --- | --- | --- | --- | --- | --- |
> > | SFT| **47.61** | 56.50 | **41.06** | 30.72 | 70.96 | 49.37 |
> > | SCOPE | 47.27 | **57.07** | 39.75 | **31.95** | **71.98** | **49.60** |
> >
> > **Q4. Effect of noisy reference labels**
> >
> > We acknowledge that datasets like XSum contain noisy gold summaries, which can negatively impact fine-tuning. However, this is precisely where SCOPE demonstrates its strength, as shown by its significant improvements over vanilla fine-tuning, validated by both automatic metrics and GPT-4-as-a-judge evaluations. While the initial tuning step of SCOPE is affected by this noise (as it relies on half the dataset), the second stage, based on relative preferences, allows the model to disprefer noisier samples. Importantly, SCOPE still learns effectively as long as the generated noisy samples are sufficiently distinct from the gold reference, which requires a properly calibrated noise level ($\alpha$). This aligns with the observations in Figure 3, where higher noise levels enhance the contrast between samples, enabling SCOPE to extract a stronger learning signal. Overall, this should make the method more robust to label noise than vanilla fine-tuning, as supported by our reference-free metrics.

---

> > ### Comment · Reviewer_hu8D · 2024-11-26
> > **Reporting confidence intervals**
> >
> > Thanks for adding additional results and improving the paper.
> >
> > Could you please report CIs for all your results? The paper often claims that the approach 'significantly improves' or 'significantly enhances', however, the significance tests or CIs are not reported.

---

> > > ### Author Response · Authors · 2024-11-27
> > >
> > > Thank you for your insightful comment regarding the reporting of statistical significance. To address your concern, we conducted statistical tests to better support our claims regarding the significance of SCOPE improvements over the SFT baseline. We have updated the manuscript to include these statistical tests, along with detailed explanations of the results. Please refer to Appendix F for more details. We summarize our findings below.
> > >
> > > **Faithfulness metrics.** We performed independent two-sample t-tests to assess whether there were statistically significant differences in the mean values of specified metrics between the baseline SFT and comparison model SCOPE. Below, you will find the p-values of the tests (also in Table 19).
> > >
> > > | Model | **ToTTo** |  | **FeTaQA** |  | **WebNLG** |  | **E2E** |  |
> > > | --- | --- | --- | --- | --- | --- | --- | --- | --- |
> > > |  | PARENT | NLI | PARENT | NLI | PARENT | NLI | PARENT | NLI |
> > > | Llama2-7b | 3.19e-50 | 4.73e-17 | 2.21e-12 | 3.68e-3 | 1.11e-31 | 4.55e-4 | 7.18e-3 | 4.01e-3 |
> > > | Llama2-13b | 4.91e-60 | 6.22e-31 | 1.37e-9 | 9.26e-2 | 1.06e-55 | 1.85e-4 | 1.48e-3 | 1.02e-2 |
> > > | Mistral-7b | 1.86e-103 | 2.26e-24 | 1.08e-3 | 1.13e-1 | 7.64e-1 | 1.68e-1 | 4.51e-5 | 2.57e-1 |
> > >
> > > |  |  | **SAMSum**  |  |  | **XSum** |  |  | **PubMed**  |  |
> > > | --- | --- | --- | --- | --- | --- | --- | --- | --- | --- |
> > > | Model | Align | FactCC | QEval | Align | FactCC | QEval | Align | FactCC | Qeval |
> > > | Llama2-7b | 1.25e-3 | 0.1077 | 4.26e-2 | 3.56e-80 | 3.54e-69 | 4.67e-144 | 3.29e-11 | 4.79e-21 | 3.47e-8 |
> > > | Llama2-13b | 1.48e-2 | 0.1216 | 3.7e-3 | 1.10e-69 | 3.27e-54 | 5.36e-160 | 1.06e-9 | 3.27e-16 | 2.53e-7 |
> > > | Mistral-7b | 3.98e-3 | 3.56e-2 | 4.38e-1 | 5.37e-73 | 3.16e-55 | 3.33e-189 | 1.20e-17 | 3.05e-22 | 1.10e-12 |
> > >
> > > Across the vast majority of datasets, metrics, and models, SCOPE is statistically significantly better than SFT (using as reference p-value < 0.05). In a small number of datasets, while most metrics show significant improvements, one or two metrics occasionally do not achieve statistical significance. However, this does not detract from the overall robustness of our findings, as the vast majority of results consistently demonstrate statistically significant improvements.
> > >
> > > **Pairwise GPT-4 evaluations.** To assess whether our SCOPE improves significantly over the other baselines based on our GPT-4 win-tie-lose pairwise preference evaluations, we perform the McNemar’s statistical test to determine if the observed difference in wins is likely due to chance or if it reflects a truly performance difference. Please refer to Appendix F for more details. Here are the p-values of the McNemar’s test on the pairwise GPT-4 evaluations:
> > >
> > > | **Comparison** | **ToTTo** | **WebNLG** | **FeTaQA** | **E2E** | **SamSum** | **XSum** | **PubMed** |
> > > | --- | --- | --- | --- | --- | --- | --- | --- |
> > > | SCOPE vs SFT | 3.70e-97 | 3.13e-23 | 9.70e-4 | 3.56e-14 | 1.171e-25 | 6.74e-153 | 3.94e-41 |
> > > | SCOPE vs PMI | 9.49e-7 | 7.70e-3 | 6.74e-1 | 4.78e-2 | 1.30e-3 | 6.27e-55 | 2.31e-19 |
> > > | SCOPE vs CRITIC | 1.47e-8 | 1.95e-2 | 2.10e-1 | 2.70e-3 | 5.41e-11 | 4.78e-74 | 2.31e-4 |
> > > | SCOPE vs CAD | 1.23e-7 | 6.79e-5 | 9.39e-2 | 1.33e-2 | 1.23e-7 | 2.61e-59 | 1.52e-11 |
> > > | SCOPE vs CLIFF | 1.23e-11 | 3.75e-11 | 6.25e-2 | 5.6e-4 | 2.04e-6 | 3.03e-04 | 1.31e-21 |
> > >
> > > The results from McNemar's test show that:
> > >
> > > **(i)** SCOPE shows consistently a significant improvement over the SFT baseline.
> > >
> > > **(ii)** most of the comparisons between SCOPE and the other baselines are statistically significant (p-value < 0.05) on ToTTo, WebNLG, E2E, SamSum, and XSum with the exception on FeTaQA.
> > >
> > > We hope these additional findings meet your expectations and further strengthen the robustness of the results.

---

> ### Author Response · Authors · 2024-12-02
>
> Dear reviewer,
>
> Thank you once again for your thoughtful and detailed feedback on our paper. As the rebuttal period concludes, we wanted to let you know that we have carefully reviewed and addressed each of your comments in our revisions. We hope the changes we made effectively address your concerns and would be grateful if you could take them into consideration during your evaluation.
>
> Thank you again for your time and effort.

---

### Author Response · Authors · 2024-11-24

We sincerely thank all reviewers for their thoughtful feedback and are grateful for the overall positive reception. In response to their suggestions, we have made several additions to the paper, including significant new experimental results:

- (Reviewer aJYu) assessments with FactCC and QuestEval, two faithfulness-focused metrics for summarization,
- (Reviewers hu8D) an evaluation of SCOPE on Alpaca, a generalist instruction-following dataset,
- (Reviewer aJYu) experiments on PubMed, a dataset for medical evidence summarization,
- (Reviewers fCdZ, aJYu) evaluations of SCOPE on a larger model (Llama2-13b),
- (Reviewers hu8D, NUJj, eSUQ) a quantitative and qualitative (human) analysis of synthetic noisy samples
- (Reviewer fCdZ) ablations on the $\beta$ hyperparameter used in preference tuning.

We are excited about these new results and hope they provide further confirmation of the effectiveness of our approach.

---

### Meta-Review · Area_Chair_B2cF · 2024-12-16

**Metareview:**

This paper introduces SCOPE, a self-supervised framework designed to improve the faithfulness of large language models (LLMs) in conditional text generation tasks, such as summarization and data-to-text generation. By leveraging a two-stage process of fine-tuning and contrastive learning, the method synthesizes negative examples (unfaithful and contextually ungrounded) and trains models to prioritize grounded outputs. Extensive evaluations, including human and GPT-4 assessments, demonstrate that SCOPE significantly outperforms existing techniques in generating faithful text, achieving notable gains across multiple benchmarks.

Strengths:

* The paper addresses significant challenges in generative models: factuality and faithfulness.
* The proposed two-stage training pipeline is both novel and well-motivated, and it is independent of external tools and resources, unlike many prior works.
* Extensive experiments across summarization and data-to-text tasks show improvements over baseline methods. The results are well-supported by a variety of metrics, human evaluations, and GPT-4 assessments, emphasizing the method's effectiveness and applicability to multiple benchmarks. The work demonstrates an ability to improve response faithfulness at a minor cost to fluency.
* The paper provides comprehensive ablation studies and analyses.

Weaknesses:
* Limited novelty: Reviewers expressed concerns about novelty, as generating artificially bad examples is not entirely new. However, the authors highlighted significant technical differences, and the simplicity of the method (requiring only a few lines of code) remains a strong point.
* Insufficient analysis: The paper lacks a detailed qualitative and quantitative analysis of the generated negative examples. However, the authors have begun to address this by including plots and analyses in the appendix.
* Scalability concerns: The experiments focus on 7B-scale models, omitting larger models where faithfulness improvements might be less pronounced. However, this focus is reasonable, as the lack of faithfulness is more apparent in smaller models. Additionally, the authors included extra experiments on 13B models, which yielded similar improvements to those observed with 7B models.
* There were other concerns, such as the fixed hyperparameter β, which were addressed during the discussion period.

After the discussion period, it appears that most major concerns have been addressed, except for the fact that there are other methods that also rely on generating artificially bad examples (though these approaches differ technically). Given the importance of the problems addressed (factuality and faithfulness in LLMs), the simplicity of the method, the positive results, and the comprehensiveness of the analysis, I recommend accepting the paper.

**Additional Comments On Reviewer Discussion:**

The paper was extensively discussed, and the authors mostly addressed the weaknesses mentioned in the meta-review. They also clarified the technical differences between their approach and related methods that rely on artificially generated bad examples.

During the reviewer-AC discussion, most reviewers supported accepting the paper, given the importance of the task and the contributions of this work (e.g., positive results and other key findings). I side with their recommendation.

---

### Decision · Program_Chairs · 2025-01-22

Accept (Poster)